# Reinforcement Learning in Linear MDPs: Constant Regret and Representation Selection

**Matteo Papini**
Universitat Pompeu Fabra
matteo.papini@upf.edu

**Andrea Tirinzoni**
INRIA Lille
andrea.tirinzoni@inria.fr

**Aldo Pacchiano**[*]
Microsoft Research
apacchiano@microsoft.com

**Marcello Restilli**
Politecnico di Milano
marcello.restelli@polimi.it

**Alessandro Lazaric**
Facebook AI Research
lazaric@fb.com

**Matteo Pirotta**
Facebook AI Research
pirotta@fb.com

## Abstract

We study the role of the representation of state-action value functions in regret minimization in finite-horizon Markov Decision Processes (MDPs) with linear structure. We first derive a necessary condition on the representation, called universally spanning optimal features (UNISOFT), to achieve constant regret in any MDP with linear reward function. This result encompasses the well-known settings of low-rank MDPs and, more generally, zero inherent Bellman error (also known as the Bellman closure assumption). We then demonstrate that this condition is also sufficient for these classes of problems by deriving a constant regret bound for two optimistic algorithms (LSVI-UCB and ELEANOR). Finally, we propose an algorithm for representation selection and we prove that it achieves constant regret when one of the given representations, or a suitable combination of them, satisfies the UNISOFT condition.

## 1 Introduction

The ability of an agent to learn an informative mapping from complex observations to a succinct representation is one of the essential factors for the success of machine learning in fields such as computer vision, language modeling, and more broadly in deep learning [Bengio et al., 2013]. In supervised learning, it is well understood that a "good" representation is one that allows to accurately fit any target function of interest (e.g., correctly classify a set of objects in an image). In Reinforcement Learning (RL), this concept is more subtle, as it can be applied to different aspects of the problem, such as the optimal value function or the optimal policy. Furthermore, recent works have shown that realizability (e.g., being able to represent the optimal value function) is not a sufficient condition for solving an RL problem, as the sample complexity using realizable representations is exponential in the worst case [e.g., Weisz et al., 2021]. As such, a desirable property of a "good" representation in RL is to enable learning a near-optimal policy with a polynomial sample complexity (or similarly sublinear regret bound).

Several works have focused on online learning — considering sample complexity or regret minimization — and identified sufficient assumptions for efficient learning. Standard examples are

---

[*]Work done while at Facebook AI Research.

35th Conference on Neural Information Processing Systems (NeurIPS 2021).

tabular Markov Decision Processes (MDPs) [e.g., Jaksch et al., 2010, Azar et al., 2012, 2017], low or zero inherent Bellman error [e.g., Jin et al., 2020, Zanette et al., 2020b,a, Jin et al., 2021] and linear mixture MDPs [e.g., Yang and Wang, 2019, Ayoub et al., 2020, Zhang et al., 2021]. While, in these settings, the representation is provided as input to the algorithm, an alternative scenario is to learn such representations. In this case, research has focused either on the problem of online representation selection for regret minimization [e.g., Ortner et al., 2014, 2019, Lee et al., 2021] or, more recently, on the sample complexity of online representation learning [e.g., Du et al., 2019, Agarwal et al., 2020, Modi et al., 2021]. Refer to App. A for more details. While this literature has focused on finding a representation enabling learning a near-optimal policy with sublinear regret or polynomial sample complexity, there may be several of such "good" representations with significantly different learning performance and existing approaches are not guaranteed to find the most efficient one. Intuitively, we would like to find representations that require the minimum level of exploration to solve the task. For example, representations that would allow the algorithm to stop exploring after a finite time and play only optimal actions forever (i.e., achieving constant regret), if they exist. This aspect of representation learning was recently studied by Hao et al. [2020], Papini et al. [2021] in contextual linear bandits, where they showed that certain representations display non-trivial properties that enable much better learning performance. While it is well-known that properties such as dimensionality and norm of the features have an impact on the learning performance, Hao et al. [2020], Papini et al. [2021] proved that it is possible to achieve constant regret (i.e., not scaling with the number of learning steps) if a certain (necessary and sufficient) condition on the features associated with the optimal actions is satisfied. To the best of our knowledge, the impact of similar properties on RL algorithms and how to find such representations is largely unexplored.

**Contributions.** In this paper, we investigate the concept of "good" representations in the context of regret minimization in finite-horizon MDPs with linear structure. In particular, we consider the settings of zero inherent Bellman error (also referred to as Bellman closure) [Zanette et al., 2020b] and low-rank structure [e.g., Jin et al., 2020]. Similarly to the bandit case [Hao et al., 2020, Papini et al., 2021], we study the impact of representations on the learning process. Our contributions are both fundamental and algorithmic. **1)** We provide a necessary condition (called UNISOFT) for a representation to enable constant regret in any problem with linear reward parametrization. Notably, this result encompasses MDPs with zero inherent Bellman error, and linear mixture MDPs with linearly parametrized rewards. Intuitively, the condition generalizes a similar condition for linear contextual bandits and it requires that the features observed along trajectories generated by the optimal actions provide information on the whole feature space (see Asm. 4). **2)** We provide the first *constant regret bound* for MDPs for both ELEANOR [Zanette et al., 2020b] and LSVI-UCB [Jin et al., 2020] when the UNISOFT condition is satisfied. As a consequence, we show that good representations are not only necessary but also sufficient for constant regret in MDPs with zero inherent Bellman error or low-rank assumptions. **3)** We develop an algorithm, called LSVI-LEADER, for representation selection in low-rank MDPs. We prove that in low-rank MDPs, LSVI-LEADER suffers the regret of the best representation without knowing it in advance. Furthermore, LSVI-LEADER achieves constant regret even when only a suitable combination of the representations satisfies the UNISOFT condition despite none of them being "good". This is indeed possible thanks to its ability to select a different representation for each stage, state, and action.

## 2 Preliminaries

We consider a time-inhomogeneous finite-horizon Markov decision process (MDP) $M = \left( \mathcal{S}, \mathcal{A}, H, \{r_h\}_{h=1}^H, \{p_h\}_{h=1}^H, \mu \right)$ where $\mathcal{S}$ is the state space and $\mathcal{A}$ is the action space, $H$ is the length of the episode, $\{r_h\}$ and $\{p_h\}$ are reward functions and state-transition probability measures, and $\mu$ is the initial state distribution. We denote by $r_h(s, a)$ the expected reward of a pair $(s, a) \in \mathcal{S} \times \mathcal{A}$ at stage $h$. We assume that $\mathcal{S}$ is a measurable space with a possibly infinite number of elements and $\mathcal{A}$ is a finite set. A policy $\pi = (\pi_1, \dots, \pi_H) \in \Pi$ is a sequence of decision rules $\pi_h : \mathcal{S} \rightarrow \mathcal{A}$. For every $h \in [H] := \{1, \dots, H\}$ and $(s, a) \in \mathcal{S} \times \mathcal{A}$, we define the value functions of a policy $\pi$ as

$$Q_h^\pi(s, a) = r_h(s, a) + \mathbb{E}_\pi \left[ \sum_{i=h+1}^H r_i(s_i, a_i) \right], \qquad V_h^\pi(s, a) = Q_h^\pi(s, \pi_h(s)),$$

where the expectation is over probability measures induced by the policy and the MDP over state-action sequences of length $H - h$. Under certain regularity conditions [e.g., Bertsekas and Shreve,

2004], there always exists an optimal policy $\pi^\star$ whose value functions are defined by $V_h^{\pi^\star}(s) := V_h^\star(s) = \sup_\pi V_h^\pi(s)$ and $Q_h^{\pi^\star}(s,a) := Q_h^\star(s,a) = \sup_\pi Q_h^\pi(s,a)$. The optimal Bellman equation (and Bellman operator $L_h$) at stage $h \in [H]$ is defined as:

$$Q_h^\star(s,a) := L_h Q_{h+1}^\star(s,a) = r_h(s,a) + \mathbb{E}_{s' \sim p_h(s,a)} \left[ \max_{a'} Q_{h+1}^\star(s',a') \right].$$

The value iteration algorithm (a.k.a. backward induction) computes $Q^\star$ or $Q^\pi$ by applying the Bellman equations starting from stage $H$ down to 1, with $V_{H+1}^\pi(s) = 0$ by definition for any $s$ and $\pi$. The optimal policy is simply the greedy policy w.r.t. $Q^\star$: $\pi_h^\star(s) = \operatorname{argmax}_{a \in \mathcal{A}} Q_h^\star(s,a)$.

In online learning, the agent interacts with an *unknown* MDP in a sequence of $K$ episodes. At each episode $k$, the agent observes an initial state $s_1^k$, it selects a policy $\pi_k$, it collects the samples observed along a trajectory obtained by executing $\pi_k$, it updates the policy, and reiterates over the next episode. We evaluate the performance of a learning agent through the regret: $R(K) := \sum_{k=1}^K V_1^\star(s_1^k) - V_1^{\pi_k}(s_1^k)$.

**Linear Representation.** When the state space is large or continuous, value functions are often described through a parametric representation. A standard approach is to use linear representations of the state-action function $Q_h(s,a) = \phi_h(s,a)^\mathsf{T} \theta_h$, where $\phi_h : \mathcal{S} \times \mathcal{A} \to \mathbb{R}^d$ is a time-inhomogeneous feature map and $\theta_h \in \mathbb{R}^d$ is an unknown parameter vector.[2] In this paper, we consider MDPs satisfying Bellman closure (i.e., zero Inherent Bellman Error) [Zanette et al., 2020b] or low-rank assumptions [e.g., Yang and Wang, 2019, Jin et al., 2020].

**Assumption 1** (Bellman Closure). *Define the set of bounded value function* $\mathcal{Q}_h = \{Q_h | \theta_h \in \Theta_h : Q_h(s,a) = \phi_h(s,a)^\mathsf{T} \theta_h, \forall (s,a)\}$ *and the associated parameter space* $\Theta_h = \{\theta_h \in \mathbb{R}^d : |\phi_h(s,a)^\mathsf{T} \theta_h| \leq D\}$. *An MDP has zero Inherent Bellman Error (IBE) if*

$$\forall h \in [H], \qquad \sup_{Q_{h+1} \in \mathcal{Q}_{h+1}} \inf_{Q_h \in \mathcal{Q}_h} \|Q_h - L_h Q_{h+1}\|_\infty = 0.$$

This definition implies that the optimal value function is realizable as $Q_h^\star \in \mathcal{Q}_h$. Furthermore, the function space $\mathcal{Q}$ is closed under the Bellman operator, i.e., for all $Q_{h+1} \in \mathcal{Q}_{h+1}$, $L_h Q_{h+1} \in \mathcal{Q}_h$. Under this assumption, value-iteration-based algorithms are guaranteed to converge to the optimal policy in the limit of samples and iterations [Munos and Szepesvári, 2008]. In the context of regret minimization, Zanette et al. [2020b] proposed a model-free algorithm, called ELEANOR, that achieves sublinear regret under the Bellman closure assumption, but at the cost of computational intractability.[3] The design of a tractable algorithm for regret minimization under low IBE assumption is still an open question in the literature.

**Assumption 2** (Low-Rank MDP). *Let* $\Theta_h = \mathbb{R}^d$, *then an MDP has low-rank structure if*

$$\forall s,a,h,s', \quad r_h(s,a) = \phi_h(s,a)^\mathsf{T} \theta_h, \quad p_h(s'|s,a) = \phi_h(s,a)^\mathsf{T} \mu_h(s')$$

*where* $\mu_h : \mathcal{S} \to \mathbb{R}^d$. *Then, for any policy* $\pi \in \Pi$, $\exists \theta_h^\pi \in \Theta_h$ *such that* $Q_h^\pi(s,a) = \phi_h(s,a)^\mathsf{T} \theta_h^\pi$. *We assume* $\|\theta_h\|_2 \leq \sqrt{d}$, $\|\int_{s'} \mu_h(s')v(s')\mathrm{d}s'\|_2 \leq \sqrt{d}\|v\|_\infty$ *and* $\|\phi_h(s,a)\|_2 \leq 1$, *for any* $s,a,h$, *and function* $v : \mathcal{S} \to \mathbb{R}$.

This assumption is *strictly* stronger than Bellman closure [Zanette et al., 2020b] and it implies the value function of *any* policy is linear in the features. Furthermore, under Asm. 2 sublinear regret is achievable using, e.g., LSVI-UCB [Jin et al., 2020], a tractable algorithm for low-rank MDPs. He et al. [2020] have recently established a problem-dependent logarithmic regret bound for LSVI-UCB under a strictly-positive minimum gap. The minimum positive gap provides a natural measure of the difficulty of an MDP.

**Assumption 3.** *The suboptimality gap of taking action* $a$ *in state* $s$ *at stage* $h$ *is defined as:*

$$\Delta_h(s,a) = V_h^\star(s) - Q_h^\star(s,a). \tag{1}$$

*We assume the minimum positive gap* $\Delta_{\min} = \min_{s,a,h}\{\Delta_h(s,a)|\Delta_h(s,a) > 0\}$ *is well defined and that the optimal action is unique, i.e.,* $|\operatorname{argmax}_a\{Q_h^\star(s,a)\}| = 1$, *for any* $s \in \mathcal{S}$, $h \in [H]$.

---

[2]It is possible to extend the setting to different feature dimensions $\{d_h\}_{h \in [H]}$.

[3]ELEANOR works under the weaker assumption of low IBE. Jin et al. [2021] considered the more general case of low Bellman Eluder dimension. Their algorithm reduces to ELEANOR in the case of low IBE.

| Algorithm (setting) | Minimax | Problem-Dependent Logarithmic | Constant with UNISOFT (this work) |
|---|---|---|---|
| ELEANOR (Bellman Closure) | $\widetilde{O}(\sqrt{d^2 H^3 T})$ [Zanette et al., 2020b] | N/A | $\frac{d^2 H^4}{\Delta_{\min}\lambda_+^{3/2}}\log^{1/2}\left(\frac{d^2 H^5}{\delta\Delta_{\min}^2\lambda_+^3}\right)$ (Thm. 8) |
| LSVI-UCB (low-rank MDPs) | $\widetilde{O}(\sqrt{d^3 H^3 T})$ [Jin et al., 2020] | $O(\frac{d^3 H^5}{\Delta_{\min}}\log^2(T))$ [He et al., 2020] | $\frac{d^3 H^5}{\Delta_{\min}}\log\left(\frac{d^4 H^6}{\delta\Delta_{\min}^2\lambda_+^3}\right)$ (Thm. 9) |
| Lower Bound | $\Omega(\sqrt{d^2 H^2 T})$ [Zhou et al., 2020, Remark 5.8] | $\Omega(\frac{dH}{\Delta_{\min}})$ [He et al., 2020] | N/A |

Table 1: Regret comparisons of ELEANOR and LSVI-UCB. For ELEANOR, we consider the special case of Bellman closure.

In Tab. 1, we summarize existing bounds in the two settings. Another structural assumption that has gained popularity in the literature is the linear-mixture structure [Jia et al., 2020, Ayoub et al., 2020, Zhou et al., 2020], where the transition function admits a form $p_h(s'|s,a) = \phi_h(s'|s,a)^\mathsf{T}\theta_h$. No structural requirement is made on the reward, which is typically assumed to be known. As a consequence, the value function may not be linearly representable. However, the fact the reward is known and that it is possible to directly learn the parameters $\theta_h$ of the transition function allow to achieve sublinear regret (even logarithmic) through model-based algorithms. While in this paper we mostly focus on Asm. 1 and 2, in Sect. 3.1 we show that our condition is necessary for constant regret also for linear-mixture MDPs with unknown linear reward.

## 3 Constant Regret for Linear MDPs

In this section, we introduce UNISOFT, a necessary condition for constant regret in any MDP with linear rewards. We show that this condition is also sufficient in MDPs with Bellman closure.

**Assumption 4.** *A feature map is* UNISOFT *(Universally Spanning Optimal FeaTures) for an MDP if it satisfies Asm. 1 or 2, and for all $h \in [H]$ the following holds:*

$$\text{span}\Big\{\phi_h(s,a) \mid \forall(s,a),\ \exists\pi \in \Pi : \rho_h^\pi(s,a) > 0\Big\} = \text{span}\Big\{\phi_h^\star(s) \mid \forall s,\ \rho_h^\star(s) > 0\Big\},$$

*where $\rho_h^\pi(s) = \mathbb{E}[\mathbb{1}\{s_h = s\}|M,\pi]$ is the occupancy measure of a policy $\pi$, $\rho_h^\pi(s,a) = \rho_h(s)\mathbb{1}\{\pi_h(s) = a\}$, $\rho_h^\star(s) := \rho_h^{\pi^\star}(s)$, and $\phi_h^\star(s) := \phi_h(s,\pi_h^\star(s))$.*

Intuitively, features that are observed by only playing optimal actions must provide information on the whole space of reachable features at each stage $h$. We notice that Asm. 4 reduces to the HLS property for contextual bandits considered by Hao et al. [2020], Papini et al. [2021]. The key difference is that, in RL, the reachability of a state plays a fundamental role. For example, features of states that are not reachable by any policy are irrelevant, while features of optimal actions in states that are not reachable by the optimal policy (i.e., $\phi_h^\star(s)$ in a state with $\rho_h^\star(s) = 0$) do not contribute to the span of optimal features since they can only be reached by acting sub-optimally. In RL, a related structural assumption to Asm. 4 is the "uniformly excited feature" assumption used by Abbasi-Yadkori et al. [2019, Asm. A4] for average reward problems. Their assumption is strictly stronger than ours since it requires that all policies generate an occupancy measure under which the features span all directions uniformly well. Such an assumption can be related to the ergodicity assumption for tabular MDPs, which is known to be restrictive. Another related quantity is the "explorability" coefficient introduced by Zanette et al. [2020c]. This term represents how explorative (in the feature space) are the optimal policies of the tasks compatible with the MDP, i.e., considering any possible parameter $\theta_h \in \Theta_h$. This coefficient is important in reward-free exploration where the objective is to learn a near optimal policy for any task, which is revealed only once learning has completed. In our setting, we focus only on the properties of the optimal policy for the single task we aim to solve.

It is interesting to look into Asm. 4 from an alternative perspective. Denote by $0 \leq \lambda_{h,1} \leq \ldots \leq \lambda_{h,d}$ the eigenvalues of the matrix $\Lambda_h := \mathbb{E}_{s\sim\rho_h^\star}\big[\phi_h^\star(s)\phi_h^\star(s)^\mathsf{T}\big]$ and by $\lambda_h^+ = \min\{\lambda_{h,i} > 0, i \in [d]\}$ the minimum positive eigenvalue. We notice that when the features are non-redundant (i.e., $\{\phi_h(s,a)\}$

spans $\mathbb{R}^d$) and the UNISOFT assumption holds, then $\lambda_h^+ = \lambda_{h,1} > 0$. As we will see, the minimum positive eigenvalue $\lambda_h^+$ plays a fundamental role in the constant regret bound, together with the minimum gap $\Delta_{\min}$. We provide examples of UNISOFT and Non-UNISOFT representations in App. G, as well as their impact on the learning process.

### 3.1 UNISOFT is Necessary for Constant Regret

The following theorem shows that the UNISOFT condition is necessary to achieve constant regret in a large class of MDPs.

**Theorem 5.** *Let $M$ be any MDP with finite states, arbitrary dynamics $p$, linear rewards (i.e., $r_h(s,a) = \phi_h(s,a)^\mathsf{T}\theta_h$) with Gaussian $\mathcal{N}(0,1)$ noise, unique optimal policy $\pi^\star$, and where condition UNISOFT (Asm. 4) is not satisfied. Let $\mathcal{M}$ be the set of MDPs with same dynamics as $M$ but different reward parameters $\{\theta_h\}_{h \in [H]}$. Then, there exists no algorithm that suffers sub-linear regret in all MDPs in $\mathcal{M}$ while suffering constant regret in $M$.*

Thm. 5 states that in MDPs with linear reward, the UNISOFT condition is *necessary* to achieve constant regret for any "provably efficient" algorithm. Notably, this result does not put any restriction on the transition model, which can be arbitrary and known. This means that as soon as the reward is linear and unknown to the learning agent, the UNISOFT condition is necessary to attain constant regret. This result applies to low-rank MDPs, linear-mixture MDPs with unknown linear rewards, and MDPs with Bellman closure (Bellman closure implies linear rewards, see Prop. 2 by Zanette et al. [2020b]).

**Proof sketch of Theorem 5.** The key intuition behind the proof is that an algorithm achieving a constant regret must select sub-optimal actions only a finite number of times. Nonetheless, in order to learn the optimal policy, all features associated with suboptimal actions should be explored enough. Since UNISOFT does not hold, this cannot happen by executing the optimal policy alone and requires selecting suboptimal policies for long enough, thus preventing constant regret.

More formally, we call an algorithm "provably efficient" if it suffers sub-linear regret on the given class of MDPs $\mathcal{M}$. Formally, we use the following definition, which is standard to prove problem-dependent lower bounds [e.g., Simchowitz and Jamieson, 2019, Xu et al., 2021].

**Definition 6** ($\alpha$-consistency). *Let $\alpha \in (0,1)$, then an algorithm A is $\alpha$-consistent on a class of MDPs $\mathcal{M}$ if, for each $M \in \mathcal{M}$ and $K \geq 1$, there exists a constant $c_M$ (independent from $K$) such that $\mathbb{E}_M^{\mathsf{A}}[R(K)] \leq c_M K^\alpha$.*[4]

The following lemma is the key result for proving Thm. 5 and it might be of independent interest. It shows that any consistent algorithm must explore sufficiently all relevant directions in the feature space to discriminate any sub-optimal policy from the optimal one. The proof (reported in App. C) leverages techniques for deriving asymptotic lower bounds for linear contextual bandits [e.g., Lattimore and Szepesvari, 2017, Hao et al., 2020, Tirinzoni et al., 2020].

**Lemma 7.** *Let $M, \mathcal{M}$ be as in Thm. 5 and A be any $\alpha$-consistent algorithm on $\mathcal{M}$. For any $\pi \in \Pi$, denote by $\Psi_h^\pi := \sum_{s,a} \rho_h^\pi(s,a)\phi_h(s,a)$ its expected features at stage $h$ and $\Delta(\pi) := V_1^\star - V_1^\pi$ its sub-optimality gap. Then, for any $\pi \in \Pi$ with $\Delta(\pi) > 0$ and $h \in [H]$,*

$$\limsup_{K \to \infty} \log(K) \|\Psi_h^\pi - \Psi_h^\star\|_{\mathbb{E}_M^{\mathsf{A}}[\Lambda_h^K]^{-1}}^2 \leq \frac{\Delta(\pi)^2}{2(1-\alpha)},$$

*where $\Psi_h^\star := \Psi_h^{\pi^\star}$ and $\Lambda_h^K := \sum_{k=1}^K \phi_h(s_h^k, a_h^k)\phi_h(s_h^k, a_h^k)^\mathsf{T}$.*

We now proceed by contradiction: suppose that A suffers constant expected regret on $M$ even though the MDP does not satisfy the UNISOFT condition. Then, since A plays sub-optimal actions only a finite number of times, it is possible to show that, for each $h \in [H]$, there exists a positive constant $\lambda_M > 0$ such that $\mathbb{E}_M^{\mathsf{A}}[\Lambda_h^K] \preceq \Lambda_h^\star + \lambda_M I$, where $\Lambda_h^\star := K \sum_{s:\rho_h^\star(s)>0} \phi_h^\star(s)\phi_h^\star(s)^\mathsf{T}$. Furthermore, since UNISOFT does not hold, there exists a stage $h \in [H]$ and a sub-optimal policy $\pi$ (i.e., with $\Delta(\pi) > 0$)

---

[4]In practice, all existing "provably-efficient" algorithms we are interested in are included in this class and $c_M$ is polynomial in all problem-dependent quantities (e.g., $d$, $H$). For instance, LSVI-UCB and ELEANOR are $1/2$-consistent on the class of low-rank and Bellman-closure MDPs, where they enjoy worst-case $\widetilde{O}(\sqrt{K})$ regret bounds (with $c_M$ being $O(\sqrt{d^3 H^4})$ and $O(\sqrt{d^2 H^4})$, respectively).

such that the vector $\Psi_h^\pi - \Psi_h^\star$ does not belong to $\text{span}\{\phi_h^\star(s)|\rho_h^\star(s) > 0\}$. Then, since such space is exactly the one spanned by all the eigenvectors of $\Lambda_h^\star$ associated with a non-zero eigenvalue, there exists a positive constant $\epsilon > 0$ (independent of $K$) such that $\|\Psi_h^\pi - \Psi_h^\star\|_{(\Lambda_h^\star + \lambda_M I)^{-1}}^2 \geq \epsilon^2/\lambda_M$. Combining these steps with Lem. 7, we obtain

$$\frac{\Delta(\pi)^2}{2(1-\alpha)} \geq \limsup_{K\to\infty} \log(K)\|\Psi_h^\pi - \Psi_h^\star\|_{(\Lambda_h^\star + \eta I)^{-1}}^2 \geq \frac{\epsilon^2}{\lambda_M} \limsup_{K\to\infty} \log(K),$$

which is clearly a contradiction. Therefore, A cannot suffer constant regret in $M$ while suffering sub-linear regret in all other MDPs in $\mathcal{M}$, and our claim follows.

### 3.2 UNISOFT is Sufficient for Constant Regret

While the UNISOFT condition is necessary for achieving constant regret in a large class of MDPs, in the following, we prove that ELEANOR and LSVI-UCB attain constant regret when the UNISOFT assumption holds, thus implying that it is a sufficient condition in MDPs with low-rank and Bellman closure structure.

**Theorem 8.** *Consider an MDP and a representation $\{\phi_h\}_{h\in[H]}$ satisfying the Bellman closure (Asm. 1) and UNISOFT assumptions (Asm. 4). Under Asm. 3, with probability at least $1 - 3\delta$, ELEANOR[5] suffers a* constant *regret*

$$R(K) \lesssim H^{3/2}d\sqrt{\overline{\tau}\log\frac{\overline{\tau}}{\delta}},$$

*where $\overline{\tau} = H\overline{\kappa}$ and $\overline{\kappa}$ is the last episode ELEANOR suffers a non-zero regret. Furthermore, $\overline{\kappa} \lesssim \max\left\{\frac{d^2 H^4}{\lambda_+^2}, \frac{dH^4}{\Delta_{\min}^2 \lambda_+^3}\right\}$[6], where $\lambda_+ := \min_h\{\lambda_h^+\} > 0$.*

Alternatively, we can prove the following result for LSVI-UCB.

**Theorem 9.** *Consider an MDP and a representation $\{\phi_h\}_{h\in[H]}$ satisfying the low-rank (Asm. 2) and UNISOFT assumptions (Asm. 4). Under Asm. 3, with probability $1 - 3\delta$, LSVI-UCB suffers a* constant *regret*

$$R(K) \lesssim \frac{d^3 H^5}{\Delta_{\min}} \log\left(dH^2\overline{\kappa}/\delta\right),$$

*where $\overline{\kappa}$ is the last episode LSVI-UCB suffers a non-zero regret and is upper-bounded as $\overline{\kappa} \lesssim \max\left\{\frac{d^3 H^4}{\lambda_+^2}, \frac{d^2 H^4}{\Delta_{\min}^2 \lambda_+^3}\right\}$, where $\lambda_+ := \min_h\{\lambda_h^+\} > 0$.*

In both cases, $\overline{\kappa}$ is polynomial in all the problem-dependent terms and independent of the number of episodes $K$ (see Lem. 21 and 20). As a result, ELEANOR and LSVI-UCB achieves a constant regret that only depends on "static" MDP and representation characteristics, thus indicating that after a finite time the agent only executes the optimal policy. Notice also that the bounds should be read as minimum between the constant regret and the minimax regret $O(\sqrt{K})$, which may be tighter for small $K$. The main difference between the two previous bounds is that for ELEANOR we build on the anytime minimax regret bound, while for LSVI-UCB, we derive a more refined constant-regret guarantee by building on its problem-dependent bound of He et al. [2020]. Unfortunately, limiting factor for applying the analysis in [He et al., 2020] seems to be the fact that ELEANOR is not optimistic at each stage $h$ but rather only at the first stage. As such, whether ELEANOR can achieve a problem-dependent logarithmic regret based on local gaps that can be leverage to improve our analysis is an open question in the literature.

**Combined proof sketch of Thm. 8 and Thm. 9.** We provide a general proof sketch that can be instantiated to both ELEANOR and LSVI-UCB. The purpose is to illustrate what properties an algorithm must have to exploit good representations, and how this leads to constant regret. Consider a learnable feature map $\{\phi_h\}_{h\in[H]}$ and an algorithm with the following properties:

(a) Greedy w.r.t. a Q-function estimate: $\pi_h^k(s) = \arg\max_{a\in\mathcal{A}}\{\overline{Q}_h^k(s,a)\}$.

---

[5]ELEANOR and LSVI-UCB are defined up to a regularization parameter $\lambda$ that we set to $\lambda = 1$.

[6]Here $\lesssim$ hides logarithmic terms in $\lambda_+$, $H$, and $d$, but not in $K$.

(b) Global optimism: $\overline{V}_1^k(s) \geq V_1^\star(s)$ where, for all $h \geq 1$, we set $\overline{V}_h^k(s) = \max_{a \in \mathcal{A}}\{\overline{Q}_h^k(s,a)\}$.

(c) Almost local optimism: $\forall h > 1, \exists C_h \geq 0$ s.t. $\overline{Q}_h^k(s,a) + C_h\beta_k \left\|\phi_h(s,a)\right\|_{(\Lambda_h^k)^{-1}} \geq Q_h^\star(s,a)$.

(d) Confidence set: let $\Lambda_h^k = \sum_{i=1}^{k-1} \phi_h(s_h^i, a_h^i)\phi_h(s_h^i, a_h^i)^\mathsf{T} + \lambda I$ and $\beta_k \in \mathbb{R}_+$ be logarithmic in $k$, then $\overline{V}_h^k(s_h^k) - V_h^{\pi_k}(s_h^k) \leq 2\beta_k \left\|\phi_h(s_h^k, a_h^k)\right\|_{(\Lambda_h^k)^{-1}} + \mathbb{E}_{s' \sim p_h(s_h^k, a_h^k)}\left[\overline{V}_{h+1}^k(s') - V_{h+1}^{\pi_k}(s')\right]$.

These properties are verified by ELEANOR [Zanette et al., 2020b, App. C] and LSVI-UCB [Jin et al., 2020, Lem. B.4, B.5]. Note that for LSVI-UCB condition (c) is trivially verified since the algorithm is optimistic at each stage ($C_h = 0$). On the other hand, ELEANOR is only guaranteed to be optimistic at the first stage, and (c) is thus important ($C_h = 2$). First, we use existing techniques to establish an *any-time* regret bound, either worst-case or problem-dependent. We call this $g(k)$ and prove that $R(k) \leq g(k) \leq \widetilde{O}(\sqrt{k})$ for any $k$ with probability $1 - 2\delta$.

Next, we show that, under Asm. 4, the eigenvalues of the design matrix grow almost linearly, making the confidence intervals decrease at a $1/\sqrt{k}$ rate. From some algebra and a martingale argument,

$$\Lambda_h^{k+1} \succeq k\Lambda_h^\star + \lambda I - \Delta_{\min}^{-1}g(k)I - \widetilde{O}(\sqrt{k})I, \tag{2}$$

where $\Lambda_h^\star = \mathbb{E}_{s \sim \rho_h^\star}[\phi_h^\star(s)\phi_h^\star(s)^\mathsf{T}]$. The UNISOFT property ensures that the linear term is nonzero in relevant directions, while the regret bound of the algorithm makes the penalty term sublinear. Then, we show that, for any *reachable* $(s,a)$,

$$\beta_k \left\|\phi_h(s,a)\right\|_{(\Lambda_h^k)^{-1}} \leq \beta_k \frac{k - \widetilde{O}(\sqrt{k})}{(k\lambda_h^+ - \widetilde{O}(\sqrt{k}))^{3/2}} = \widetilde{O}(k^{-1/2}), \tag{3}$$

where $\lambda_h^+$ is the minimum *nonzero* eigenvalue of $\Lambda_h^\star$. From (3), we can see that $\lambda_h^+$ plays a fundamental role in the rate of decrease. Finally, we show that, under the gap assumption, these uniformly-decreasing confidence intervals allow learning the optimal policy in a finite time. From the Bellman equations, we have that

$$V_1^\star(s_1^k) - V_1^{\pi^k}(s_1^k) = \mathbb{E}_{\pi^k}\left[\sum_{h=1}^H \Delta_h(s_h, a_h)|s_1 = s_1^k\right], \tag{4}$$

while from (a)-(d), for any reachable state,

$$\Delta_h(s, \pi_h^k(s)) \leq 2\,\mathbb{E}_{\pi^k}\left[\sum_{i=h}^H \beta_k \left\|\phi_i(s_i, a_i)\right\|_{(\Lambda_i^k)^{-1}} |s_h = s\right] + \mathbb{1}_{h>1}C_h\beta_k \left\|\phi_h^\star(s)\right\|_{(\Lambda_h^k)^{-1}}.$$

The second term (with $\mathbb{1}_{h>1}$) accounts for the almost-optimism of ELEANOR, while it is zero in LSVI-UCB due to the stage-wise optimism. Then, for every $h \in [H]$, we can use (3) to control the feature norms. Thus, there exists an episode $\kappa_h$ independent of $K$ satisfying

$$\Delta_h(s, \pi_h^k(s)) \leq \beta_{\kappa_h} \sum_{i=h}^H (2 + \mathbb{1}_{i=h>1}C_h) \frac{\kappa_h - 8\sqrt{\kappa_h \log(2d\kappa_h H/\delta)} - g(\kappa_h)}{(\kappa_h\lambda_i^+ - 8\sqrt{\kappa_h \log(2d\kappa_h H/\delta)} - g(\kappa_h))^{3/2}} < \Delta_{\min}, \tag{5}$$

By definition of minimum gap, then $\Delta_h(s, \pi_h^k(s)) = 0$ for $k > \kappa_h$. Then, for $k > \overline{\kappa} = \max_h\{\kappa_h\}$, $V_1^\star(s_1^k) - V_1^{\pi^k}(s_1^k) = 0$. But this means the algorithm only accumulates regret up to $\overline{\kappa}$, that is, $R(K) = R(\overline{\kappa}) \leq g(\overline{\kappa}) = O(1)$ for all $K > \overline{\kappa}$. This holds with probability $1 - 3\delta$, also taking into account the martingale argument from (2). Note that $\{\kappa_h\}$ are by definition monotone for LSVI-UCB. The final bounds are then obtained by instantiating the specific values of $\beta_k$ and $g(k)$ for the two algorithms we analyzed.

## 4 Representation Selection in Low-Rank MDPs

In Sec. 3, we have highlighted the benefits that a UNISOFT representation brings to optimistic algorithms in MDPs with Bellman closure and low rank structure. In this section, we take one step further and investigate the *representation selection* problem. Since ELEANOR is a computationally

**Algorithm 1:** LSVI-LEADER

**Input:** Representations $\{\Phi_j\}_{j\in[N]}$, confidence values $\{\beta_k\}_{k\in[K]}$

1 **for** $k = 1, \ldots, K$ **do**
2  Receive the initial state $s_1^k$
3  **for** $h = H, \ldots, 1$ **do**
4   $\Lambda_h^k(j) = \lambda I + \sum_{i=1}^{k-1} \phi_h^{(j)}(s_h^i, a_h^i)\phi_h^{(j)}(s_h^i, a_h^i)^\mathsf{T}\ \forall j \in [N].$
5   $\boldsymbol{w}_h^k(j) = \Lambda_h^k(j)^{-1}\sum_{i=1}^{k-1}\phi_h^{(j)}(s_h^i, a_h^i)\left(r_h(s_h^i, a_h^i) + \max_{a\in\mathcal{A}}\overline{Q}_{h+1}^k(s_{h+1}^i, a)\right),\ \forall j\in[N]$
6   $\overline{Q}_h^k(s,a) = \min\left\{H, \min_{j\in[N]}\left(\phi_h^{(j)}(s,a)^\mathsf{T}\boldsymbol{w}_h^k(j) + \beta_k\left\|\phi_h^{(j)}(s,a)\right\|_{\Lambda_h^k(j)^{-1}}\right)\right\}$
7  **for** $h = 1, \ldots, H$ **do**
8   Execute action $a_h^k = \pi_h^k(s_h^k) := \operatorname{argmax}_{a\in\mathcal{A}}\overline{Q}_h^k(s_h^k, a).$

intractable algorithm, we build on LSVI-UCB and low-rank MDPs (Asm. 2) and we introduce LSVI-LEADER (Alg. 1), an algorithm that adaptively selects representations in a given set.

Given a set of $N$ representations $\{\Phi_j\}_{j\in[N]}$ satisfying Asm. 2, where $\Phi_j = \{\phi_h^{(j)}\}_{h\in[H]}$, at each stage $h \in [H]$ of episode $k \in [K]$, LSVI-LEADER solves $N$ different regression problems to compute an optimistic value function for each representation. Then, the final estimate $\overline{Q}_h^k(s,a)$ is taken as the *minimum* across these different optimistic value functions. Notably, this implies that LSVI-LEADER implicitly *combines* representations, in the sense that the selected representations (i.e., those with tightest optimism) might vary for different stages. This is exploited in the following result, which shows that constant regret is achievable even if none of the given representations is globally UNISOFT.

**Theorem 10.** *Given an MDP $M$ and a set of representations $\{\Phi_j\}_{j\in[N]}$ satisfying the low-rank assumption (Asm. 2), let $\mathcal{Z}$ be the set of $H^N$ representations obtained by combining those in $\{\Phi_j\}_{j\in[N]}$ across different stages.[7] Then, with probability at least $1 - 2\delta$, LSVI-LEADER suffers at most a regret*

$$R(K) \leq \min_{z\in\mathcal{Z}}\widetilde{R}(K, z, \{\beta_k\}),$$

*where $\widetilde{R}(K, z, \beta_k)$ is either the worst-case regret bound of LSVI-UCB [Jin et al., 2020] or the problem-dependent one [He et al., 2020] when the algorithm is executed with representation $z$ and confidence values $\beta_k \propto dH\sqrt{N\log(2dNHk/\delta)}$. Moreover, if $\mathcal{Z}$ contains a UNISOFT representation $z^\star$, then LSVI-LEADER achieves constant regret with problem-dependent values of $z^\star$ (see Thm. 9).*

This result shows that LSVI-LEADER adapts to the *best* representation automatically, i.e., without any prior knowledge about the properties of the representations. In particular, it shows a problem-dependent (or worst-case) bound when there is no UNISOFT representation, while it attains constant regret when a representation, potentially mixed through stages, is UNISOFT. This is similar to what was obtained by Papini et al. [2021] for linear contextual bandits. Indeed, LSVI-LEADER reduces to their algorithm in the case $H = 1$. While the cost of representation selection is only logarithmic in linear bandits, the cost becomes polynomial (i.e., $\sqrt{N}$ in the worst-case bound and $N$ in the problem-dependent one) in RL. This is due to the structure induced by the Bellman equation, which requires a cover argument over $H^N$ functions (more details in the proof sketch). Note that for $H = 1$, the analysis can be refined to obtain a $\log(N)$ dependence, due to the lack of propagation through stages, and recover the result in [Papini et al., 2021]. We refer the read to App. G for a numerical validation.

**Proof sketch of Thm. 10.** The proof relies on the following important result, which extends Lem. B.4 of Jin et al. [2020] and shows that the deviation between the optimistic value function computed by LSVI-LEADER and the true one scales with the *minimum* confidence interval across the different representations. Formally, with probability $1 - 2\delta$, for any $\pi \in \Pi, s \in \mathcal{S}, a \in \mathcal{A}, h \in$

---

[7]Note that any combination of features in $\Phi_j$ is learnable, since each representation is learnable in the low-rank MDP sense.

$[H], k \in [K]$,

$$\overline{Q}_h^k(s,a) - Q_h^\pi(s,a) \le 2\beta_k \min_{j \in [N]} \left\| \phi_h^{(j)}(s,a) \right\|_{\Lambda_h^k(j)^{-1}} + \mathbb{E}_{s' \sim p_h(s,a)} \left[ \overline{V}_{h+1}^k(s') - V_{h+1}^\pi(s') \right].$$

As in [Jin et al., 2020], the derivation of this result combines the well-known self-normalized martingale bound in [Abbasi-Yadkori et al., 2011] with a covering argument over the space of possible optimistic value functions. In our setting, the structure of such function space requires us to build $N$ different covers, one for each different representation. This, in turn, requires the confidence values $\beta_k$ to be inflated by an extra factor $\sqrt{N}$ w.r.t. learning with a single representation.

The generality of this result allows us to easily derive, for any fixed representation $z \in \mathcal{Z}$, both the worst-case regret bound of Jin et al. [2020] and the problem-dependent one of He et al. [2020]. To see this, note that the regret decompositions in both of these two papers rely on an upper bound to $\overline{V}_h^k(s_h^k) - V_h^{\pi_k}(s_h^k)$ as a function of the *fixed* representation used by LSVI-UCB (see the proof of Theorem 3.1 of Jin et al. [2020] and Lemma 6.2 of He et al. [2020]). Then, fix any $z \in \mathcal{Z}$ and call $z_h$ its features at stage $h$. Note that $z_h \in \{\phi_h^{(j)}\}_{j \in [M]}$. Moreover, by definition of low-rank structure, since each $\Phi_j$ induces a low-rank MDP, their combination does too. Thus, $z$ is learnable. Then, instantiating the concentration bound stated above for policy $\pi^k$, state $s_h^k$, action $a_h^k$, stage $h$, and by upper bounding the minimum with the representation selected in $z_h$, we get

$$\overline{V}_h^k(s_h^k) - V_h^{\pi_k}(s_h^k) \le 2\beta_k \left\| z_h(s_h^k, a_h^k) \right\|_{\Lambda_h^k(j)^{-1}} + \mathbb{E}_{s' \sim p_h(s_h^k, a_h^k)} \left[ \overline{V}_{h+1}^k(s') - V_{h+1}^{\pi_k}(s') \right].$$

From here, one can carry out exactly the same proofs of Jin et al. [2020] and He et al. [2020], thus obtaining the same regret bound that LSVI-UCB enjoys when executed with the fixed representation $z \in \mathcal{Z}$ and confidence values $\{\beta_k\}_{k \in [K]}$. Hence, we conclude that the regret of LSVI-LEADER is upper bounded by the minimum of these regret bounds for all representations $z \in \mathcal{Z}$, thus proving the first result. To obtain the second result, simply notice that, if $z^\star \in \mathcal{Z}$ is UNISOFT, then we can use the refined analysis for LSVI-UCB of Thm. 9 to show that $\widetilde{R}(K, z^\star, \{\beta_k\})$ is upper bounded by a constant independent of $K$, hence proving constant regret for LSVI-LEADER.

## 4.1 Representation Selection under a Mixing Condition

We show that the LSVI-LEADER algorithm not only is able to select the best representation among a set of viable representations, and to combine representations for the different stages, but also to stitch representations together *across states and actions*. With this in mind we introduce the notion of a mixed ensemble of representations.

**Definition 11.** *Consider an MDP $M$ and a set of representations $\{\Phi_j\}_{j \in [N]}$ satisfying the low-rank assumption (Asm. 2). The collection of feature maps $\{\Phi_j\}_{j \in [M]}$ is* UNISOFT-*mixing if for all* $s, a \in \mathcal{S} \times \mathcal{A}$ *and* $h \in [H]$, *there exists $j$ such that $\phi_h^{(j)}(s,a) \in \mathrm{span}\left\{\phi_h^{(j)}(s, \pi_h^\star(s)) | \rho_h^\star(s) > 0\right\}$.*

We show that when presented with a UNISOFT-mixing family of representations, LSVI-LEADER is able to successfully combine these and obtain a regret guarantee that may be better than what is achievable by running LSVI-UCB using any of these representations in isolation.

**Theorem 12.** *Consider an MDP $M$ and a set of representations $\{\Phi_j\}_{j \in [N]}$ satisfying the low-rank (Asm. 2) and* UNISOFT-*mixing assumptions. If $\Delta_{\min} > 0$ (Asm. 3), then with probability at least $1 - 3\delta$, there exist a constant $\widetilde{\kappa} = \max_h \{\kappa_h\}$ independent from $K$ such that the regret of LSVI-LEADER after $K$ episodes is at most:*

$$R(K) \le \min_{z \in \mathcal{Z}} \widetilde{R}(\widetilde{\kappa}, z, \{\beta_k\}),$$

*where $\mathcal{Z}$, $\widetilde{R}$ and $\beta_k$ are defined as in Thm. 10.*

Under the UNISOFT-mixing condition, LSVI-LEADER may not converge to selecting a single representation for each stage $h$ but rather to mixing multiple representations. In fact, it may select a different representation in different regions of the state-action space. This is the main difference w.r.t. Thm. 10, where constant regret is shown when there exists a representation $z^\star$ that is UNISOFT, and the value $\kappa_h$ depends on the minimum positive eigenvalue of $z_h^\star$. In the case of UNISOFT-mixing, $\kappa_h$ depends on properties of a combination of representations at stage $h$. We provide a characterization of $\kappa_h$ in the full proof in App. E.

# 5 Conclusions

We investigated the properties that make a representation efficient for online learning in MDPs with Bellman closure. We introduced UNISOFT, a necessary and sufficient condition to achieve a constant regret bound in this class of MDPs. We demonstrate that existing optimistic algorithms are able to adapt to the structure of the problem and achieve constant regret. Furthermore, we introduce an algorithm able to achieve constant regret by mixing representations across states, actions and stages in the case of low-rank MDPs. An interesting direction raised by our paper is whether it is possible to leverage the UNISOFT structure for probably-efficient representation learning, rather than selection. Another direction can be to leverage these insights to drive the design of auxiliary losses for representation learning, for example in deep RL.

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
