# A  Related Work

The representation selection problem has been originally studied in the context of tabular MDPs. Given a set of representation mapping histories to (sequences of actions, observations, and rewards) to a finite set of states, the goal of the learning agent is to solve the MDP under an appropriate representation. The standard assumption is that at least one representation induces an MDP. Several papers have investigated this online learning problem and provided algorithms based on the optimism principle [e.g., Ortner et al., 2014, 2019]. The settings and the representation learning objective are different from ours. In particular, this line of research aims at finding any representation that is good for learning but the methods are not guaranteed to find the most efficient.

Recently, a few papers have focused on representation learning with theoretical guarantees. Du et al. [2019] considered the representation learning problem in block MDPs with rich observations, where the objective is to learn the compact latent representation. Representation learning in low-rank MDPs was recently studied in [Agarwal et al., 2020, Modi et al., 2021, Lu et al., 2021]. We believe that these papers are orthogonal to our work for several reasons. We start considering the setting in [Agarwal et al., 2020, Modi et al., 2021]. First, they operate in the reward-free setting where the objective is to learn a representation of the low-rank MDP that can be used to efficiently learn an optimal policy once a reward is given. For us, a reward is given from the start and learning/selecting a good representation in the meantime is just a way to suffer less regret. Second, our representation selection objective is different (and arguably more challenging) than the one considered by Agarwal et al. [2020], Modi et al. [2021]. They aim at finding a representation with low mean square error, i.e. any realizable representation of the low-rank MDP. On the other hand, we wish to find a UNISOFT representation among a set of realizable representations, which makes the representation learning problem harder. In App. F, Papini et al. [2021] showed that reducing the MSE is not enough for this purpose. It is shown that this only allows the algorithm to end up with a set of realizable representations, but after that, a different algorithmic scheme, whose primary objective is reducing regret (like LSVI-LEADER), is needed to find the UNISOFT one. Therefore, even if the approach in these papers could be extended to the regret minimization setting, there would be no guarantee that running that algorithm would recover a UNISOFT representation as in our case. Finally, it is unclear how to transform their sample complexity into a regret bound. In particular, it is not just a matter of translating a sample complexity bound into a regret bound: both the interaction protocol and the algorithmic schemes are different w.r.t. our work. Even if we directly translated the sample complexity bounds of these papers into regret bounds, we note that, while it is true that they could scale as $\log(|\Phi|)$, they also contain several dependencies which are orders of magnitude worse than in our work. For instance, the sample complexity provided in [Modi et al., 2021, Thm. 2] scales as $A^{13}$ ($A$ is the number of actions). This is also an unreasonable dependence in any case of practical interest we can think of. Finally, Lu et al. [2021] studied the effect of representation learning on the sample complexity in multi-task settings, which is quite different from the single-task regret minimization problem considered in this paper.

# B  Notation

Table 2: Notation.

| | | |
|---|---|---|
| $\mathcal{S}$ | | state space |
| $\mathcal{A}$ | | action space |
| $H$ | | episode length |
| $r_h$ | | reward function at stage $h$ |
| $p_h$ | | transition function at stage $h$ |
| $\mu$ | | initial-state distribution |
| $K$ | | number of episodes |
| $T$ | $=$ | $HK$, total number of interactions |
| $\pi_h$ | | policy for stage $h$ |
| $\Pi$ | | policy space |
| $Q_h^\pi$ | | state-action value function of policy $\pi$ at stage $h$ |
| $V_h^\pi$ | $=$ | $Q_h^\pi(s, \pi_h(s))$ |
| $\pi_h^\star$ | | optimal policy for stage $h$ |
| $Q_h^\star$ | $=$ | $Q_h^{\pi^\star}$, optimal value function at stage $h$ |
| $V_h^\star(s)$ | $=$ | $\max_{a \in \mathcal{A}} Q_h^\star(s, a)$ |
| $L_h$ | | Bellman's optimality operator for stage $h$ |
| $\pi_h^k$ | | policy played by the algorithm at stage $h$ of episode $k$ |
| $\phi_h$ | | feature map for stage $h$ |
| $R(K)$ | | regret suffered in the first $K$ episodes |
| $d$ | | feature dimension |
| $D$ | $=$ | $H$, value function upper bound |
| $\mathcal{Q}_h$ | | set of linear bounded value functions for stage $h$ |
| $\Theta_h$ | | set of parameters of linear bounded value functions for stage $h$ |
| $\Delta_h(s, a)$ | $=$ | $V_h^\star(s) - Q_h^\star(s, a)$, suboptimality gap |
| $\Delta_{\min}$ | | minimum positive gap (see Asm. 3) |
| $\phi_h^\star(s)$ | $=$ | $\phi_h(s, \pi_h^\star(s))$, optimal features for state $s$ at stage $h$ |
| $\rho_h^\pi$ | | occupancy measure of policy $\pi$ at stage $h$ (see Asm. 4) |
| $\Lambda_h^\star$ | $=$ | $\mathbb{E}_{s \sim \rho_h^\star}[\phi_h^\star(s)\phi_h^\star(s)]$, optimal covariance matrix |
| $\lambda_h^+$ | | minimum nonzero eigenvalue of $\Lambda_h^\star$ |
| $\delta$ | | failure probability |
| $\overline{\kappa}$ | | last episode at which nonzero regret is paid (see proof of Thm. 19) |
| $\overline{\tau}$ | $=$ | $H\overline{\kappa}$ |
| $\beta_k$ | | confidence radius, see (32) for ELEANOR and (42) for LSVI-UCB |
| $\lambda$ | $=$ | 1, regularization parameter |
| $\Lambda_h^k$ | $=$ | $\lambda I + \sum_{i=1}^{k-1} \phi_h(s_h^i, a_h^i)\phi_h(s_h^i, a_h^i)^\mathsf{T}$, design matrix |
| $\overline{Q}_h^k$ | | optimistic value function for stage $h$ at episode $k$ |
| $\overline{V}_h^k(s)$ | $=$ | $\max_{a \in \mathcal{A}} \overline{Q}_h^k(s, a)$ |

## C   UNISOFT is Necessary: Proofs of Section 3.1

We illustrate all the detailed proofs needed for showing that the UNISOFT condition is necessary to achieve constant regret (Thm. 5). For the sake of completeness, we restate here all the assumptions on the MDP $M$ under consideration.

**Assumptions on MDP $M$.**

- $\mathcal{S}$ and $\mathcal{A}$ finite, $H \geq 1$ arbitrary;
- Linear rewards: $r_h(s,a) = \langle \theta_h, \phi(s,a) \rangle$ with $\mathcal{N}(0,1)$ noise;
- Arbitrary transition probabilities $\{p_h\}_{h \in [H]}$ and initial-state distribution $\mu$;
- Unique optimal policy $\pi^\star$: $|\{a : Q_h^\star(s,a) = V_h^\star(s)\}| = 1$ and $\pi_h^\star(s) = \operatorname{argmax}_a Q_h^\star(s,a)$ for all $s, h$;
- UNISOFT condition (Asm. 4 does not hold).

Moreover, recall that we define $\mathcal{M}$ as any set of MDPs that contains (but it can be larger than) all the MDPs which are equivalent to $M$ in all components except for the reward parameters $\{\theta_h\}_{h \in [H]}$, which can be arbitrary vectors in $\mathbb{R}^d$. Formally,

$$\mathcal{M} \supseteq \left\{ \widetilde{M} = \left( \mathcal{S}, \mathcal{A}, H, \{\widetilde{r}_h\}_{h=1}^H, \{p_h\}_{h=1}^H, \mu \right) \mid \forall h \in [H], \exists \widetilde{\theta}_h \in \mathbb{R}^d : \widetilde{r}_h(s,a) = \langle \widetilde{\theta}_h, \phi(s,a) \rangle \right\}.$$

Intuitively, $\mathcal{M}$ contains at least all the MDPs that could be faced by an agent that knows the linear-reward structure of the problem but that does not know the true parameters $\{\theta_h\}_{h \in [H]}$. Obviously, if the agent knows all the components of $M$ except for the reward parameters, the set $\mathcal{M}$ can be taken exactly as the set on the righthand side above (which would contain all and only the realizable MDPs). On the other hand, in the more general case where the agent does not know the dynamics as well, set $\mathcal{M}$ can be enlarged by including all the realizable MDPs with different transition probabilities (e.g., those with low-rank or low-IBE structure, or even the whole set of unstructured dynamics). Our proof that UNISOFT is necessary for constant regret holds for an agent that only knows that the true MDP $M$ belongs to this general set $\mathcal{M}$ and thus encompasses all the relevant settings mentioned in Sec. 3.1.

In the following proofs we shall write $\mathbb{P}_M^{\mathsf{A}}$ ($\mathbb{E}_M^{\mathsf{A}}$) to denote the probability (expectation) operator under MDP $M$ and the chosen algorithm A.

### C.1   Proof of Lemma 7

Let $M$ be our true MDP and $\widetilde{M} \in \mathcal{M}$ be any other MDP which is equivalent to $M$ in all components except for the reward parameters, which are given by $\{\widetilde{\theta}_h\}_{h \in [H]}$. We start by a standard decomposition of the expected log-likelihood ratio between the observations generated in the two MDPs. Fix $K \geq 1$ and let $\mathrm{KL}(\mathbb{P}_M, \mathbb{P}_{\widetilde{M}})$ denote the KL-divergence between the distributions of the observations collected by algorithm A over $K$ episodes. Using, e.g., Lemma 5 of Domingues et al. [2021] together with the closed-form of the KL divergence between Gaussian distributions,

$$\mathrm{KL}(\mathbb{P}_M, \mathbb{P}_{\widetilde{M}}) = \sum_{s,a} \sum_{h \in [H]} \mathbb{E}_M^{\mathsf{A}}[N_h^K(s,a)] \frac{(\langle \phi(s,a), \theta_h - \widetilde{\theta}_h \rangle)^2}{2} = \frac{1}{2} \sum_{h \in [H]} \|\theta_h - \widetilde{\theta}_h\|_{\mathbb{E}_M^{\mathsf{A}}[\Lambda_h^K]}^2,$$

where $\Lambda_h^K := \sum_{s,a} N_h^K(s,a) \phi(s,a) \phi(s,a)^T$ and $N_h^K(s,a) := \sum_{k=1}^K \mathbb{1}\{s_h^k = s, a_h^k = a\}$.

Suppose that, for sufficiently large $K$, the matrix $\mathbb{E}_M^{\mathsf{A}}[\Lambda_h^K]$ is invertible.[8] We now proceed as follows. For a fixed $h \in [H]$ and sub-optimal policy $\pi \in \Pi$ (i.e., with $\Delta(\pi) > 0$), we seek the hardest MDP $\widetilde{M}$ to discriminate from $M$ (i.e., that minimizes $\mathrm{KL}(\mathbb{P}_M, \mathbb{P}_{\widetilde{M}})$) where policy $\pi$ is strictly better (in terms of expected return) than $\pi^\star$ and where we change only the parameter $\theta_h$ w.r.t. $M$. Formally, we minimize

$$\operatorname{minimize}_{\widetilde{\theta}_h \in \mathbb{R}^d} \|\theta_h - \widetilde{\theta}_h\|_{\mathbb{E}_M^{\mathsf{A}}[\Lambda_h^K]}^2$$

---

[8]Lattimore and Szepesvari [2017] proved that this is indeed true for consistent algorithms. Otherwise, one could simply make the matrix positive-definite by adding $\lambda I$ for some arbitrary $\lambda > 0$ and the derivation still holds.

subject to the constraint $\widetilde{V}_1^\pi \geq \widetilde{V}_1^{\pi^\star} + \epsilon$. First note that the expected return of policy $\pi$ can be equivalently written as

$$V_1^\pi = \sum_{s,a} \sum_{h\in[H]} \rho_h^\pi(s,a) r_h(s,a) = \sum_{h\in[H]} \langle \theta_h, \sum_{s,a} \rho_h^\pi(s,a)\phi(s,a)\rangle = \sum_{h\in[H]} \langle \theta_h, \Psi_h^\pi \rangle.$$

Moreover, since $M$ and $\widetilde{M}$ have same transition probabilities, $\Psi_h^\pi = \widetilde{\Psi}_h^\pi$ for each $\pi, h$. Thus, $\widetilde{V}_1^\pi = \sum_{h\in[H]} \langle \widetilde{\theta}_h, \Psi_h^\pi \rangle$ and the constraint can be rewritten in the more convenient form

$$\sum_{h\in[H]} \langle \widetilde{\theta}_h, \Psi_h^\pi \rangle \geq \sum_{h\in[H]} \langle \widetilde{\theta}_h, \Psi_h^\star \rangle + \epsilon.$$

Using Lemma 13, the optimization problem has a closed-form expression. Therefore, let $\Gamma_h^\epsilon(\pi) \subseteq \mathcal{M}$ be the set of MDPs over which we are optimizing, that is, with (1) same transition probabilities as $\mathcal{M}$, (2) same reward parameters as $\mathcal{M}$ at all stages except $h$, and (3) $\widetilde{V}_1^\pi \geq \widetilde{V}_1^{\pi^\star} + \epsilon$. Using Lemma 13 together with the rewritings above, for any $\pi \in \Pi$, $h \in [H]$ and $\epsilon \geq 0$,

$$\min_{\widetilde{M}\in\Gamma_h^\epsilon(\pi)} \mathrm{KL}(\mathbb{P}_M, \mathbb{P}_{\widetilde{M}}) = \frac{(\Delta(\pi)+\epsilon)^2}{2\|\Psi_h^\pi - \Psi_h^\star\|^2_{\mathbb{E}_M^\mathsf{A}[\Lambda_h^K]^{-1}}}. \tag{6}$$

We now show that $\mathrm{KL}(\mathbb{P}_M, \mathbb{P}_{\widetilde{M}})$ is lower bounded by a quantity that increases logarithmically in $K$ for any $\widetilde{M} \in \Gamma_h^\epsilon(\pi)$ with $\epsilon > 0$. Let $E_K := \{\sum_{\pi\in\Pi^\star} N_K(\pi) < f(K)\}$, where $N_K(\pi) := \sum_{k=1}^K \mathbb{1}\{\pi^k = \pi\}$, $\Pi^\star$ is the set of all deterministic policies with maximal expected return in $M$, and $f(K)$ will be specified later. Using Lemma 14,

$$\mathrm{KL}(\mathbb{P}_M, \mathbb{P}_{\widetilde{M}}) \geq \log \frac{1}{\mathbb{P}_M(E_K) + \mathbb{P}_{\widetilde{M}}(E^c)} - \log 2. \tag{7}$$

Now note that, under the assumption that A is $\alpha$-consistent,

$$c_M K^\alpha \geq \mathbb{E}_M^\mathsf{A}[R(K)] = \sum_{\pi\in\Pi} \mathbb{E}_M^\mathsf{A}[N_K(\pi)]\Delta(\pi) \geq \Delta \sum_{\pi\notin\Pi^\star} \mathbb{E}_M^\mathsf{A}[N_K(\pi)].$$

Here, with some abuse of notation, $\Delta$ is the minimum policy gap. Therefore,

$$\mathbb{P}_M(E_K) = \mathbb{P}_M \left( K - \sum_{\pi\notin\Pi^\star} N_K(\pi) < f(K) \right) \leq \frac{\sum_{\pi\notin\Pi^\star} \mathbb{E}_M^\mathsf{A}[N_K(\pi)]}{K - f(K)} \leq \frac{K^\alpha c_M/\Delta}{K - f(K)},$$

where the first inequality is Markov's inequality. Note that, since $\Psi_h^\pi = \Psi_h^\star$ for all optimal policies $\pi \in \Pi^\star$ and since the transition probablities of $M$ and $\widetilde{M}$ are the same, $\widetilde{V}_1^\pi = \widetilde{V}_1^{\pi^\star}$ for all $\pi \in \Pi^\star$. Hence, all optimal policies for $M$ have a gap of at least $\epsilon$ in $\widetilde{M}$. This implies that

$$c_{\widetilde{M}} K^\alpha \geq \mathbb{E}_{\widetilde{M}}^\mathsf{A}[R(K)] \geq \epsilon \mathbb{E}_{\widetilde{M}}^\mathsf{A}\left[ \sum_{\pi\in\Pi^\star} N_K(\pi) \right].$$

Therefore,

$$\mathbb{P}_{\widetilde{M}}(E_K^c) = \mathbb{P}_{\widetilde{M}} \left( \sum_{\pi\in\Pi^\star} N_K(\pi) \geq f(K) \right) \leq \frac{\mathbb{E}_{\widetilde{M}}^\mathsf{A}\left[ \sum_{\pi\in\Pi^\star} N_K(\pi) \right]}{f(K)} \leq \frac{K^\alpha c_{\widetilde{M}}/\epsilon}{f(K)}.$$

If we set $f(K) = K/2$ and plug the two bounds above into (7), we obtain

$$\mathrm{KL}(\mathbb{P}_M, \mathbb{P}_{\widetilde{M}}) \geq \log \frac{K^{1-\alpha}}{2c_M/\Delta + 2c_{\widetilde{M}}/\epsilon} - \log 2.$$

Finally, for any $\widetilde{M} \in \Gamma_h^\epsilon(\pi)$ with $\epsilon > 0$,

$$\liminf_{K\to\infty} \frac{\mathrm{KL}(\mathbb{P}_M, \mathbb{P}_{\widetilde{M}})}{\log(K)} \geq 1 - \alpha.$$

This holds for any $\epsilon > 0$. Hence, in combination with (6), we proved that, for any sub-optimal policy $\pi$ and stage $h$,

$$\liminf_{K\to\infty} \frac{1}{\log(K)} \frac{\Delta(\pi)^2}{2\|\Psi_h^\pi - \Psi_h^\star\|^2_{\mathbb{E}_M^\mathsf{A}[\Lambda_h^K]^{-1}}} \geq 1 - \alpha.$$

Rearranging concludes the proof.

## C.2 Proof of Theorem 5

We now use Lemma 7 to prove that the UNISOFT condition is necessary for constant regret. We proceed in different steps.

**Step 1. Controlling the design matrix.** Suppose that the algorithm suffers constant regret on instance $M$. This means that, for some constant $C_M$ (different from the $c_M$ used in the definition of $\alpha$-consistence),

$$\mathbb{E}_M^{\mathsf{A}}\left[\mathrm{R}(K)\right] \leq C_M. \tag{8}$$

Since $\mathbb{E}_M^{\mathsf{A}}\left[\mathrm{R}(K)\right] = \sum_h \sum_{s,a} \mathbb{E}_M^{\mathsf{A}}\left[N_h^K(s,a)\right]\Delta_h(s,a)$, we have that $\sum_h \sum_{s,a\neq\pi_h^\star(s)}\mathbb{E}_M^{\mathsf{A}}\left[N_h^K(s,a)\right] \leq C_M/\Delta_{\min}$, where $\Delta_{\min}$ is the minimum value-function gap. Therefore, the expected design matrix at each $h \in [H]$ satifies

$$
\begin{aligned}
\mathbb{E}_M^{\mathsf{A}}[\Lambda_h^K] &= \sum_{s,a}\mathbb{E}_M^{\mathsf{A}}[N_h^K(s,a)]\phi(s,a)\phi(s,a)^T \\
&= \sum_s \mathbb{E}_M^{\mathsf{A}}[N_h^K(s,\phi_h^\star(s))]\phi_h^\star(s)\phi_h^\star(s)^T + \sum_{s,a\neq\pi_h^\star(s)}\mathbb{E}_M^{\mathsf{A}}[N_h^K(s,a)]\phi(s,a)\phi(s,a)^T \\
&\preceq \sum_s \mathbb{E}_M^{\mathsf{A}}[N_h^K(s)]\phi_h^\star(s)\phi_h^\star(s)^T + L^2\frac{C_M}{\Delta_{\min}}I \\
&\preceq K\sum_{s:\rho_h^\star(s)>0}\phi_h^\star(s)\phi_h^\star(s)^T + \sum_{s:\rho_h^\star(s)=0}\mathbb{E}_M^{\mathsf{A}}[N_h^K(s)]\phi_h^\star(s)\phi_h^\star(s)^T + L^2\frac{C_M}{\Delta_{\min}}I.
\end{aligned}
$$

We now bound the expected number of times the algorithm visit states which are not visited by an optimal policy. Take any $s$ such that $\rho_h^\star(s) = 0$. Since any optimal policy has the same state distribution $\rho_h^\star$, the event $s_h^k = s$ implies that $\pi^k \notin \Pi^\star$. Therefore,

$$\mathbb{E}_M^{\mathsf{A}}[N_h^K(s)] = \mathbb{E}_M^{\mathsf{A}}[\sum_{k=1}^K \mathbb{1}\left\{s_h^k = s\right\}] \leq \mathbb{E}_M^{\mathsf{A}}[\sum_{k=1}^K \mathbb{1}\left\{\pi^k \notin \Pi^\star\right\}] = \mathbb{E}_M^{\mathsf{A}}[\sum_{\pi\notin\Pi^\star}N_K(\pi)].$$

Moreover, since the algorithm suffers constant regret,

$$\Delta\mathbb{E}_M^{\mathsf{A}}[\sum_{\pi\notin\Pi^\star}N_K(\pi)] \leq \mathbb{E}_M^{\mathsf{A}}\left[\mathrm{R}(K)\right] \leq C_M.$$

Therefore, we conclude that

$$\mathbb{E}_M^{\mathsf{A}}[\Lambda_h^K] \preceq K\sum_{s:\rho_h^\star(s)>0}\phi_h^\star(s)\phi_h^\star(s)^T + L^2\left(\frac{C_M}{\Delta_{\min}} + S_h\frac{C_M}{\Delta}\right)I,$$

where $S_h := S - |\mathrm{supp}(\rho_h^\star))|$.

**Step 2. Controlling the feature expectations.** We now show that, since UNISOFT does not hold, there exists a sub-optimal policy $\pi$ such that $\Psi_h^\pi$ is not in the span of the optimal features. By directly using the definition of UNISOFT (Asm. 4), we have that there must exist a state-action pair $s, a$ which is reachable at time $h$ (i.e., $\exists \pi \in \Pi : \rho_h^\pi(s,a) > 0$) such that $\phi(s,a) \notin \mathrm{span}\left\{\phi_h^\star(s)|\rho_h^\star(s) > 0\right\}$. Clearly, we have only two cases:

1. $\rho_h^\star(s) > 0$ and $a \neq \pi_h^\star(s)$;
2. $\rho_h^\star(s) = 0$ and $a$ is arbitrary (even an optimal action).

For Case 1, simply take a policy $\pi$ that is equivalent to $\pi^\star$ everywhere except that $\pi_h(s) = a$. Clearly, the policy is sub-optimal, in the sense that $\Delta(\pi) = V_1^\star - V_1^\pi > 0$. Moreover, it is easy to check that $\Psi_h^\pi - \Psi_h^\star = \rho_h^\star(s)(\phi(s,a) - \phi_h^\star(s))$. Therefore, $\Psi_h^\pi \notin \mathrm{span}\left\{\phi_h^\star(s)|\rho_h^\star(s) > 0\right\}$.

For Case 2, choose $\pi$ in such a way that $\rho_h^\pi(s) > 0$ (we know that one such policy exists due to the reachability of $s$). This only requires selecting the actions of $\pi$ for all stages $h' < h$. For all stages

$h' > h$, set $\pi$ equal to $\pi^\star$ except for $\pi_h(s) = a$. Note that, even if $a$ is optimal at time $h$, $\pi$ is strictly sub-optimal (i.e., $\Delta(\pi) > 0$) since no optimal policy can achieve the condition $\rho_h^\pi(s) > 0$ by the uniqueness of the optimal state distribution. Moreover,

$$
\begin{aligned}
\Psi_h^\pi - \Psi_h^\star &= \sum_{s',a'} \rho_h^\pi(s',a')\phi(s',a') - \sum_{s'} \rho_h^\star(s')\phi_h^\star(s') \\
&= \rho_h^\pi(s)\phi(s,a) - \underbrace{\rho_h^\star(s)}_{=0}\phi_h^\star(s) + \sum_{s' \neq s}(\rho_h^\pi(s',a') - \rho_h^\star(s'))\phi_h^\star(s').
\end{aligned}
$$

Thus, we still conclude $\Psi_h^\pi \notin \mathrm{span}\,\{\phi_h^\star(s)|\rho_h^\star(s) > 0\}$.

**Step 3. Concluding the proof.** Combining Lemma 7 with Step 1 and Step 2, we have that, for some $h \in [H]$ and policy $\pi$ such that $\Delta(\pi) > 0$ and $\Psi_h^\pi \notin \mathrm{span}\,\{\phi_h^\star(s)|\rho_h^\star(s) > 0\}$,

$$
\limsup_{K \to \infty} \log(K)\|\Psi_h^\pi - \Psi_h^\star\|_{(\Lambda_h^\star + \eta I)^{-1}}^2 \leq \frac{\Delta(\pi)^2}{2(1-\alpha)},
$$

where $\Lambda_h^\star := K \sum_{s:\rho_h^\star(s)>0} \phi_h^\star(s)\phi_h^\star(s)^T$ and $\eta := L^2 \left( \frac{C_M}{\Delta_{\min}} + S_h \frac{C_M}{\Delta} \right) > 0$. Using Lemma 34, we have that there exists an $\epsilon > 0$ (independent of $K$) such that $\|\Psi_h^\pi - \Psi_h^\star\|_{(\Lambda_h^\star + \eta I)^{-1}} \geq \frac{\epsilon}{\sqrt{\eta}}$. Therefore, we get that

$$
\limsup_{K \to \infty} \log(K) \leq \frac{\eta\Delta(\pi)^2}{2\epsilon^2(1-\alpha)},
$$

which clearly does not hold since the left-hand side grows with $K$ while the right-hand side is constant. Therefore, we have a contradiction, and the algorithm A cannot achieve constant regret on this non-UNISOFT instance while being consistent on all other instances in $\mathcal{M}$. Our claim that UNISOFT is necessary follows.

### C.3 Auxiliary Results

**Lemma 13.** *Let $A \in \mathbb{R}^{d \times d}$ be any positive semi-definite invertible matrix. For $\pi \in \Pi$, $h \in [H]$, and $\epsilon \geq 0$, consider the following optimization problem:*

$$
\begin{aligned}
\min_{\theta \in \mathbb{R}^d} \quad & \|\theta - \theta_h\|_A^2 \\
\text{subject to} \quad & \sum_{l \in [H], l \neq h} \langle \theta_l, \Psi_l^\pi - \Psi_l^\star \rangle + \langle \theta, \Psi_h^\pi - \Psi_h^\star \rangle \geq \epsilon
\end{aligned}
$$

*Then, for $\overline{\theta}$ a minimizer we have*

$$
\|\overline{\theta} - \theta_h\|_A^2 = \frac{(\Delta(\pi) + \epsilon)^2}{\|\Psi_h^\pi - \Psi_h^\star\|_{A^{-1}}^2}.
$$

*Proof.* To simplify notation, let us define $b := \sum_{l \in [H], l \neq h} \langle \theta_l, \Psi_l^\pi - \Psi_l^\star \rangle$. The corresponding Lagrange dual problem is

$$
\max_{\lambda \geq 0} \min_{\theta \in \mathbb{R}^d} \left\{ \|\theta - \theta_h\|_A^2 - \lambda \left( \langle \theta, \Psi_h^\pi - \Psi_h^\star \rangle + b - \epsilon \right) \right\}.
$$

Let $f(\theta, \lambda)$ denote the resulting objective function. Taking the gradient w.r.t. $\theta$,

$$
\nabla_\theta f(\theta, \lambda) = 2A(\theta - \theta_h) - \lambda(\Psi_h^\pi - \Psi_h^\star),
$$

and equating it to zero, we obtain

$$
\theta = \theta_h + \frac{\lambda}{2}A^{-1}(\Psi_h^\pi - \Psi_h^\star).
$$

Plugging this back to the original objective we get

$$f(\lambda) = \frac{\lambda^2}{4} \|A^{-1}(\Psi_h^\pi - \Psi_h^\star)\|_A^2 - \lambda \left( \langle \theta_h, \Psi_h^\pi - \Psi_h^\star \rangle + \frac{\lambda}{2} \|\Psi_h^\pi - \Psi_h^\star\|_{A^{-1}}^2 + b - \epsilon \right)$$

$$= -\frac{\lambda^2}{4} \|\Psi_h^\pi - \Psi_h^\star\|_{A^{-1}}^2 - \lambda \left( \langle \theta_h, \Psi_h^\pi - \Psi_h^\star \rangle + \sum_{l \in [H], l \neq h} \langle \theta_l, \Psi_l^\pi - \Psi_l^{\pi^\star} \rangle - \epsilon \right)$$

$$= -\frac{\lambda^2}{4} \|\Psi_h^\pi - \Psi_h^\star\|_{A^{-1}}^2 + \lambda \left( \Delta(\pi) + \epsilon \right).$$

Differentiating with respect to $\lambda$ and equating to zero we obtain

$$\lambda = \frac{2\left(\Delta(\pi) + \epsilon\right)}{\|\Psi_h^\pi - \Psi_h^\star\|_{A^{-1}}^2}.$$

Therefore, plugging this back into the objective value

$$\|\overline{\theta} - \theta_h\|_A^2 = \frac{\left(\Delta(\pi) + \epsilon\right)^2}{\|\Psi_h^\pi - \Psi_h^\star\|_{A^{-1}}^2}.$$

$\square$

**Lemma 14** (Bretagnolle–Huber inequality, see, e.g., Thm. 14.2 of Lattimore and Szepesvári [2020]). *Let $\mathbb{P}$ and $\mathbb{Q}$ be probability measures on the same measurable space $(\Omega, \mathcal{F})$ and let $E \in \mathcal{F}$ be an arbitrary event. Then,*

$$\mathbb{P}(E) + \mathbb{Q}(E^c) \geq \frac{1}{2} e^{-\mathrm{KL}(\mathbb{P}, \mathbb{Q})}.$$

## D  UNISOFT is Sufficient: Proofs of Section 3.2

We first prove that UNISOFT is sufficient for a whole class of algorithms, as done in the proof sketch of Section 3.2. We will then instantiate this result to ELEANOR and LSVI-UCB.

Consider the following assumptions.

**Assumption 15.** *Consider a feature map $\{\phi_h\}_{h \in [H]}$ and a Q-function estimate $\overline{Q}_h^k$. There is an event $G(\delta)$ that holds with probability at least $1 - \delta$ under which:*

(a) *Global optimism: $\overline{V}_1^k(s) \geq V_1^\star(s)$ where $\overline{V}_h^k(s) = \max_{a \in \mathcal{A}} \{\overline{Q}_h^k(s, a)\}$,*

(b) *Confidence set: let $\Lambda_h^k = \sum_{i=1}^{k-1} \phi_h(s_h^i, a_h^i) \phi_h(s_h^i, a_h^i)^\mathsf{T} + \lambda I$ and $\beta_k \in \mathbb{R}_+$ be increasing and logarithmic in $k$, then $\overline{V}_h^k(s_h^k) - V_h^{\pi^k}(s_h^k) \leq 2\beta_k \|\phi_h(s_h^k, a_h^k)\|_{(\Lambda_h^k)^{-1}} + \mathbb{E}_{s' \sim p_h(s_h^k, a_h^k)} \left[ \overline{V}_{h+1}^k(s') - V_{h+1}^{\pi^k}(s') \right],$*

*simultaneously for all $h \in [H]$, $k \geq 1$ and $s \in \mathcal{S}$, where $\delta \in (0, 1)$ is a parameter of the algorithm.*

**Assumption 16.** *The algorithm satisfies Assumption 15, and additionally there exist a set of constants $(C_h)_{h \in [H]}$ such that, under the event $G(\delta)$:*

(c) *(Almost) local optimism:*  $\overline{Q}_h^k(s, a) + C_h \beta_k \|\phi_h(s, a)\|_{(\Lambda_h^k)^{-1}} \geq Q_h^\star(s, a),$

*for all $h = 2, \ldots, H$, $k \geq 1$, $s \in \mathcal{S}$ and $a \in \mathcal{A}$.*

Assumption 16 characterizes the class of algorithms for which we are going to prove a constant bound on the regret under UNISOFT. However, we first study the regret under the weaker Assumption 15, following the proof pattern from [Jin et al., 2020].

**Lemma 17.** *Under Assumption 15, assuming event $G(\delta)$ holds, there exists a $\widetilde{O}(\sqrt{K})$ function $g$ such that, with probability $1 - \delta$, for all $K \geq 1$:*

$$R(K) \leq H\beta_K \sqrt{2dK \log(1 + K/\lambda)} + 2H^2 \sqrt{K \log(2HK/\delta)} = \widetilde{O}(\sqrt{K}). \tag{9}$$

*Proof.* Under event $G(\delta)$:

$$R(K) = \sum_{k=1}^{K} V_1^{\star}(s_1^k) - V_1^{\pi^k}(s_1^k)$$

$$\leq \sum_{k=1}^{K} \overline{V}_1^k(s_1^k) - V_1^{\pi^k}(s_1^k) \qquad \text{(a)} \qquad (10)$$

$$\leq 2 \underbrace{\sum_{h=1}^{H} \beta_K \sum_{k=1}^{K} \left\| \phi_h(s_h^k, a_h^k) \right\|_{(\Lambda_h^k)^{-1}}}_{(A)} + \underbrace{\sum_{k=1}^{K} \sum_{h=1}^{H} \zeta_h^k}_{(B)}, \qquad (11)$$

where the last inequality is from recursive application of (b) and the fact that $\beta_k$ is increasing, and:

$$\zeta_h^k = \mathbb{E}_{s' \sim p_h(s_h^k, a_h^k)}[\overline{V}_{h+1}^k(s') - V_{h+1}^{\pi^k}(s')] - \overline{V}_{h+1}^k(s_{h+1}^k) + V_{h+1}^{\pi^k}(s_{h+1}^k), \qquad (12)$$

where expectations are conditioned on the history up to the beginning of episode $k$. We bound $(A)$ using the Elliptical Potential Lemma [e.g., Abbasi-Yadkori et al., 2011]:

$$(A) = 2\beta_K \sum_{h=1}^{H} \sum_{k=1}^{K} \left\| \phi_h(s_h^k, a_h^k) \right\|_{(\Lambda_h^k)^{-1}} \qquad (13)$$

$$2\beta_K \sum_{h=1}^{H} \leq \sqrt{K \sum_{k=1}^{K} \left\| \phi_h(s_h^k, a_h^k) \right\|_{(\Lambda_h^k)^{-1}}^2} \qquad (14)$$

$$\leq H\beta_K \sqrt{2dK \log(1 + K/\lambda)}. \qquad (15)$$

Since $\zeta_h^k$ is a martingale difference sequence with $\zeta_h^k \leq 2H$, we can use Azuma's inequality (Prop. 27) to bound $(B)$:

$$\sum_{k=1}^{K} \zeta_h^k \leq 2H\sqrt{K \log(2K/\delta_h)}, \qquad (16)$$

with probability $1 - \delta_h$ for all $K \geq 1$. To make it hold with probability $1 - \delta$ for all $h \in [H]$, we set $\delta_h = \delta/H$. Finally:

$$(B) = \sum_{h=1}^{H} \sum_{k=1}^{K} \zeta_h^k \leq 2H^2 \sqrt{K \log(2HK/\delta)}. \qquad (17)$$

$\square$

The stronger Assumption 16 is needed to upper-bound the gaps.

**Lemma 18.** *Under Assumption 16, assuming event $G(\delta)$ holds, for all $s \in \mathcal{S}$, $h \in [H]$ and $k \geq 1$:*

$$\Delta_h(s, \pi_h^k(s)) \leq 2\, \mathbb{E}_{\pi^k} \left[ \sum_{i=h}^{H} \beta_k \left\| \phi_i(s_i, a_i) \right\|_{(\Lambda_i^k)^{-1}} \,\middle|\, s_h = s \right] + \mathbb{1}\,\{h > 1\}\, C_h \beta_k \left\| \phi^{\star}(s) \right\|_{(\Lambda_h^k)^{-1}}.$$

*Proof.*

$$\Delta_h(s, \pi_h^k(s)) = V_h^\star(s) - Q_h^\star(s_h^k, \pi_h^k(s)) \tag{18}$$

$$\leq V_h^\star(s) - Q_h^{\pi^k}(s_h^k, \pi_h^k(s)) \tag{19}$$

$$= V_h^\star(s) - V_h^{\pi^k}(s) \tag{20}$$

$$= Q_h^\star(s, \pi_h^\star(s)) - V_h^{\pi^k}(s) \tag{21}$$

$$\leq \overline{Q}_h^k(s, \pi_h^\star(s)) + \mathbb{1}\{h > 1\} C_h \beta_k \|\phi_h(s, \pi_h^\star(s))\|_{(\Lambda_h^k)^{-1}} - V_h^{\pi^k}(s) \tag{22}$$

$$\leq \overline{V}_h^k(s) + \mathbb{1}\{h > 1\} C\sqrt{\gamma_{hk}} \|\phi_h(s, \pi_h^\star(s))\|_{\Lambda_{hk}^{-1}} - V_h^{\pi^k}(s) \tag{23}$$

$$\leq 2\,\mathbb{E}_{\pi^k}\left[\sum_{i=h}^H \beta_k \|\phi_i(s_i, a_i)\|_{(\Lambda_i^k)^{-1}} \,\middle|\, s_h = s\right] \tag{24}$$
$$+ \mathbb{1}\{h > 1\} C_h \beta_k \|\phi_h(s_h^k, \pi_h^\star(s_h^k))\|_{(\Lambda_h^k)^{-1}},$$

where (22) uses (a) for $h = 1$ and (c) for $h > 1$, while the last inequality is from recursive application of (b). $\qquad\square$

Now we can prove our main result on constant regret:

**Theorem 19.** *Any algorithm satisfying Assumption 16 enjoys constant regret if the representation has the UNISOFT property (Asm. 4) and Assumption 3 on the minimum gap holds. In general, let $g : \mathbb{N} \to \mathbb{R}_+$ be any increasing $\widetilde{O}(\sqrt{K})$ function such that, with probability $1 - 2\delta$ for all $K \geq 1$, $R(K) \leq g(K)$. Then, under Assumptions 3, 4, 16, with probability $1 - 3\delta$ for all $K \geq 1$:*

$$R(K) \leq g(\overline{\kappa}) = O(1), \tag{25}$$

*where $\overline{\kappa}$ is a constant independent of $K$.*

*Proof.* First notice that a valid regret upper bound $g(K)$ always exists due to Lemma 17. Moreover, due to Asm. 4, for all $h \in [H]$ and $k \geq 1$, we have $\phi_h(s, \pi_h^k(s)) \in \mathrm{span}\{\phi_h^\star(s) | \rho_h^\star(s) > 0\}$ for all $s \in \mathcal{S}$ such that $\rho_h^{\pi^k}(s) > 0$. Hence, with probability $1 - 2\delta$, the requirements of Lemma 33 are satisfied and we can apply it to the gap upper bound from Lemma 18. So, with probability $1 - 3\delta$, for all $s \in \mathcal{S}, h \in [H]$ and $k \geq \widetilde{\kappa} = \max_{h \in [H]} \widetilde{\kappa}_h$:

$$\Delta_h(s, \pi_h^k(s)) \leq 2\,\mathbb{E}_{\pi^k}\left[\sum_{i=h}^H \beta_k \|\phi_i(s_i, a_i)\|_{(\Lambda_i^k)^{-1}} \,\middle|\, s_h = s\right]$$
$$+ \mathbb{1}\{h > 1\} C_h \beta_k \|\phi^\star(s)\|_{(\Lambda_h^k)^{-1}} \tag{26}$$

$$\leq (2 + \mathbb{1}\{h > 1\} C_h) \beta_k \sum_{i=h}^H \frac{k + \lambda - g(k) - 8\sqrt{k \log(2dHk/\delta)}}{(k\lambda_i^+ + \lambda - g(k) - 8\sqrt{k \log(2dHk/\delta)})^{3/2}}. \tag{27}$$

Assume for now that $k \geq \widetilde{\kappa}$. From the previous inequality, since $g(k) = \widetilde{O}(\sqrt{k})$ and $\beta_k = \widetilde{O}(1)$, there exists a $\kappa_h$ independent of $K$ such that, for $k > \kappa_h$:

$$\Delta_h(s, \pi_h^k(s)) \leq \Delta_{\min}. \tag{28}$$

Under Asm. 3, this implies $\Delta_h(s, \pi_h^k(s)) = 0$. Let $\overline{\kappa} = \max\{\widetilde{\kappa}, \max_h\{\kappa_h\}\}$. For $k > \overline{\kappa}$, all the gaps are zero. Finally, by Prop. 29:

$$R(K) = \sum_{k=1}^K \mathbb{E}_{\pi^k}\left[\sum_{h=1}^H \Delta_h(s_h, a_h) \,\middle|\, s_1 = s_1^k\right] \tag{29}$$

$$= \sum_{k=1}^{\overline{\kappa}} \mathbb{E}_{\pi^k}\left[\sum_{h=1}^H \Delta_h(s_h, a_h) \,\middle|\, s_1 = s_1^k\right] + \sum_{k=\overline{\kappa}+1}^K \mathbb{E}_{\pi^k}\left[\sum_{h=1}^H \underbrace{\Delta_h(s_h, a_h)}_{=0} \,\middle|\, s_1 = s_1^k\right] \tag{30}$$

$$= R(\overline{\kappa}) \leq g(\overline{\kappa}). \tag{31}$$

$\qquad\square$

Finally, we instantiate the general result of 19 to ELEANOR on MDPs with Bellman closure and LSVI-UCB on low-rank MDPs, by showing that they satisfy Assumption 16.

**Proof of Theorem 8.**

Let:

$$\beta_k = H\sqrt{\frac{d}{2}\log(1+k/d)+d\log(1+4\sqrt{dk})+\log\frac{2Hk^2}{\delta}}+1, \tag{32}$$

and define event $G(\delta)$ as in Lemma 2 from [Zanette et al., 2020b]. We have (a) by Lemma 7 from [Zanette et al., 2020b], while (b) can be extracted from the proof of Theorem 1 from [Zanette et al., 2020b]. To prove (c), we use the fact that the MDP satisfies Bellman closure, hence there exist $\theta_1^\star, \ldots, \theta_H^\star$ such that [Lemma 6 from Zanette et al., 2020b]:

$$Q_h^\star(s,a) = \phi_h(s,a)^\mathsf{T}\theta_h^\star. \tag{33}$$

By Lemma 7 from [Zanette et al., 2020b], $\theta_1^\star, \ldots, \theta_H^\star$ is a feasible solution for $\overline{\theta}_1, \ldots, \overline{\theta}_H$ in ELEANOR's program [Definition 2 from Zanette et al., 2020b]. Due to the program's constraints:

$$\left\|\theta_h^\star - \widehat{\theta}_h^k\right\|_{\Lambda_h^k} \le \beta_k. \tag{34}$$

Let $\overline{\theta}_1^k, \ldots, \overline{\theta}_H^k$ be the values that are actually selected by ELEANOR's program. Since they are subject to the same constraints, by the triangular inequality:

$$\left\|\theta_h^\star - \overline{\theta}_h^k\right\|_{\Lambda_h^k} \le 2\beta_k. \tag{35}$$

Finally, since $\overline{Q}_h^k(s,a) = \phi_h(s,a)^\mathsf{T}\overline{\theta}_h^k$:

$$Q_h^\star(s_h, a_h) = \phi_h(s_h, a_h)^\mathsf{T}\theta_h^\star \tag{36}$$

$$= \phi_h(s_h, a_h)^\mathsf{T}\overline{\theta}_h^k + \phi_h(s_h, a_h)^\mathsf{T}(\theta_h^\star - \overline{\theta}_h^k) \tag{37}$$

$$\le \overline{Q}_h^k(s_h, a_h) + \|\phi(s_h, a_h)\|_{(\Lambda_h^k)^{-1}}\left\|\theta_h^\star - \overline{\theta}_h^k\right\|_{\Lambda_h^k} \tag{38}$$

$$\le \overline{Q}_h^k(s_h, a_h) + 2\beta_k \|\phi(s_h, a_h)\|_{(\Lambda_h^k)^{-1}}, \tag{39}$$

so (c) holds with $C_h = 2$. So Asm. 16 holds and we can invoke Theorem 19 with the upper bound $g$ from Lemma 17 and the $\beta_k$ given above to obtain:

$$R(K) \le H^2\left(\sqrt{\frac{d}{2}\log(1+\overline{\kappa}/d)+d\log(1+4\sqrt{d\overline{\kappa}})+\log(H\overline{\kappa}^2)+\log\frac{2}{\delta}}+H\right)$$

$$\times \sqrt{2d\overline{\kappa}\log(1+\overline{\kappa}/\lambda)}+2H^2\sqrt{\overline{\kappa}\log(2H\overline{\kappa}/\delta)} \tag{40}$$

$$\lesssim H^{3/2}d\sqrt{\overline{\tau}\log\frac{\overline{\tau}}{\delta}}, \tag{41}$$

where $\overline{\tau} = H\overline{\kappa}$. $\qquad\square$

**Remark 1.** We have slightly modified the ELEANOR algorithm to obtain any-time regret bounds. In particular, we have replaced the fixed $\delta' = \delta/(2T)$ term in the original $\beta_k$ (see the proof of Lemma 2 in [Zanette et al., 2020b]) with the adaptive $\delta/(2Hk^2)$. This still makes event $G(\delta)$ hold with probability $1-\delta$, but without knowledge of the horizon $K$. This only affects logarithmic terms. Also notice that we have considered the case of zero inherent Bellman error ($\mathcal{I} = 0$), which corresponds to Bellman closure, and we have taken $[0, H]$, not $[0, 1]$, as the range of the value function (see the comment following Theorem 1 in [Zanette et al., 2020b]).

For LSVI-UCB, we can instantiate Theorem 19 with the problem-dependent logarithmic lower bound by He et al. [2020] in place of the worst-case upper bound from Lemma 17.

**Proof of Theorem 9.**

Let:

$$\beta_k = c_\beta dH \sqrt{\log(2dHk/\delta)}, \tag{42}$$

where $c_\beta$ is a constant defined in Lemma C.3 from [Jin et al., 2020], and define event $G(\delta)$ as in Lemma B.3 from [Jin et al., 2020]. Then since the MDP is low-rank, by Lemma B.5 from [Jin et al., 2020] we have both (a) and (c) with $C_h = 0$. We get (b) by Lemma B.4 from Jin et al. [2020]. So Asm. 16 holds and, under Asm 3, we can instantiate Theorem 19 with the logarithmic regret bound from Theorem 4.4 by He et al. [2020]:

$$g(k) = 9HG(k)\log G(k) + \frac{16H^2}{3}\log\frac{\log\lceil Hk\rceil}{\delta} + 2, \tag{43}$$

where:

$$G(k) \propto \frac{d^3H^4\log(4dH^2k(k+1)\log(H/\Delta_{\min})/\delta)}{\Delta_{\min}}. \tag{44}$$

So:

$$R(K) \leq g(\overline{\kappa}) \simeq \frac{d^3H^5}{\Delta_{\min}}\log\left(dH^2\overline{\kappa}/\delta\right). \tag{45}$$

$\square$

**Remark 2.** We have slightly modified the LSVI-UCB algorithm to obtain any-time regret bounds. In particular, we have replaced the fixed $\iota = \log(2dT/\delta)$ term in the original $\beta_k$ (see Theorem 3.1 from [Jin et al., 2020]) with the adaptive $\log(4dHk^2/\delta)$. This still makes event $G(\delta)$ hold with probability $1 - \delta$, but without knowledge of the horizon $K$. We have also re-written the logarithmic regret bound by He et al. [2020] (Theorem 4.4) to hold with probability $1 - 2\delta$. These changes only affect logarithmic terms.

**Lemma 20.** *The critical time $\overline{\kappa}$ from Theorem 9 for LSVI-UCB is upper bounded as:*

$$\overline{\kappa} \leq \max\left\{\frac{48c_1^2H^4d^3}{\lambda_+^2}\log\left(\frac{32c_1^2H^5d^4}{\lambda_+^2\delta}\right), \frac{432c_2^2H^4d^2}{\Delta_{\min}^2\lambda_+^3}\log\left(\frac{288d^3H^5c_2^2}{\Delta_{\min}^2\lambda_+^3\delta}\right)\right\} \tag{46}$$

*where $\lambda_+ = \min_{h\in[H]}\{\lambda_h^+\}$ and $c_1, c_2$ are universal constants.*

*Proof.* For LSVI-UCB we have (see the proof of Theorem 9):

$$g(k) \leq c_1H^2d^{3/2}\sqrt{k\log(2dHk/\delta)}, \tag{47}$$

$$\beta_k = c_2dH\sqrt{\log(2dHk/\delta)}, \tag{48}$$

for some universal constants $c_1, c_2$. We assume $\lambda = 1$ and $c_1 \geq 8$.

We will use the fact that a sufficient condition for $k \geq a\log(bk)$ is $k \geq 3a\log(ab)$ for $k > 0$ and reasonable values of the constants $a, b$. See App. C.6 from Papini et al. [2021] for details. This immediately implies that a sufficient condition for $k \geq a\sqrt{k\log(bk)}$ is:

$$k \geq 3a^2\log(a^2b) \tag{49}$$

We divide the rest of the proof in three parts:

**Part 1.** First, $\kappa$ must satisfy the invertibility conditions from Lemma 33. To make matrix $B_h^k = k\Lambda_h^\star + \lambda I - g(k) + 8\sqrt{k\log(2dHk/\delta)}$ invertible for each $h$, we first require the positive eigenvalues of $\Lambda_h^\star$ to map into positive eigenvalues of $B_h^k$. A sufficient condition for this is:

$$k\lambda_+ > 1 + g(k) + 8\sqrt{k\log(2dHk/\delta)} \tag{50}$$

$$k \geq \frac{c_1H^2d^{3/2} + 8}{\lambda_+}\sqrt{k\log(2dHk/\delta)} \tag{51}$$

$$k \geq \frac{2c_1H^2d^{3/2}}{\lambda_+}\sqrt{k\log(2dHk/\delta)} \tag{52}$$

$$k \geq \frac{12c_1^2H^4d^3}{\lambda_+^2}\log\left(\frac{8c_1^2H^5d^4}{\lambda_+^2\delta}\right) \triangleq \overline{\kappa}_1, \tag{53}$$

where the latter is from (49). We also need the zero eigenvalues of $\Lambda_h^\star$ to map into negative eigenvalues of $B_h^k$. However, this just requires $\lambda - g(k) + 8\sqrt{k\log(2dHk/\delta)} < 0$ which is already true for $k = 1$ given $\lambda = 1$.

**Part 2.** We require $\overline{\kappa}$ to satisfy the following, which will make the analysis of Part 3 easier:

$$g(k) + 8\sqrt{k\log(2dHk/\delta)} \leq \frac{k\lambda_+}{2}. \tag{54}$$

$\square$

After rearranging, we can proceed precisely as in Part 1, only with different numerical constants, obtaining:

$$k \geq \frac{48c_1^2 H^4 d^3}{\lambda_+^2} \log\left(\frac{32c_1^2 H^5 d^4}{\lambda_+^2 \delta}\right) \triangleq \overline{\kappa}_2. \tag{55}$$

**Part 3.** Assume for now that $k \geq \overline{\kappa}_2$. Since $\overline{\kappa}_2 \geq \overline{\kappa}_1$, the invertibility conditions from Lemma 33 are satisfied and, by the proof of Theorem 19, regret is zero for all time $k$ such that:

$$(2 + \mathbb{1}\{h > 1\} C_h)\beta_k \sum_{i=h}^{H} \frac{k + \lambda - g(k) - 8\sqrt{k\log(2dHk/\delta)}}{(k\lambda_i^+ + \lambda - g(k) - 8\sqrt{k\log(2dHk/\delta)})^{3/2}} \leq \Delta_{\min}, \tag{56}$$

for all $h$. Using the definition of $\lambda^+$, $\lambda = 1$ and $C_h = 0$ for LSVI-UCB, a sufficient condition is:

$$2H\beta_k \frac{k + 1 - g(k) - 8\sqrt{k\log(2dHk/\delta)}}{(k\lambda_+ + 1 - g(k) - 8\sqrt{k\log(2dHk/\delta)})^{3/2}} \leq \Delta_{\min}, \tag{57}$$

$$2H\beta_k \frac{2k}{(k\lambda_+ - g(k) - 8\sqrt{k\log(2dHk/\delta)})^{3/2}} \leq \Delta_{\min}. \tag{58}$$

Since $k \geq \overline{\kappa}_2$, by (54), we just need:

$$2H\beta_k \frac{2k}{\left(\frac{1}{2}k\lambda_+\right)^{3/2}} \leq \Delta_{\min}. \tag{59}$$

Rearranging and using the definition of $\beta_k$:

$$\sqrt{k} \geq \frac{12c_2 H^2 d}{\Delta_{\min}\lambda_+^{3/2}}\sqrt{\log(2dHk/\delta)} \tag{60}$$

$$k \geq \frac{12c_2 H^2 d}{\Delta_{\min}\lambda_+^{3/2}}\sqrt{k\log(2dHk/\delta)}, \tag{61}$$

and again from (49):

$$k \geq \frac{432c_2^2 H^4 d^2}{\Delta_{\min}^2 \lambda_+^3} \log\left(\frac{288c_2^2 d^3 H^5}{\Delta_{\min}^2 \lambda_+^2 \delta}\right) \triangleq \overline{\kappa}_3. \tag{62}$$

The proof is concluded by taking $\overline{\kappa} = \max\{\overline{\kappa}_2, \overline{\kappa}_3\}$.

**Lemma 21.** *The critical time $\overline{\kappa}$ from Theorem 8 for ELEANOR is upper bounded as:*

$$\overline{\kappa} \leq \max\left\{\frac{48c_1^2 H^4 d^2}{\lambda_+^2} \log\left(\frac{32c_1^2 H^5 d^3}{\lambda_+^2 \delta}\right), \frac{432c_2^2 H^4 d}{\Delta_{\min}^2 \lambda_+^3} \log\left(\frac{288d^2 H^5 c_2^2}{\Delta_{\min}^2 \lambda_+^3 \delta}\right)\right\} \tag{63}$$

*where $\lambda_+ = \min_{h \in [H]}\{\lambda_h^+\}$ and $c_1, c_2$ are universal constants.*

*Proof.* The proof is the same as for Lemma 20, except that for ELEANOR we have (see the proof of Theorem 8):

$$g(k) \leq c_1 H^2 d\sqrt{k\log(2dHk/\delta)} \tag{64}$$

$$\beta_k \leq c_2 H\sqrt{d\log(2dHk/\delta)}, \tag{65}$$

where $c_1, c_2$ are universal constants. The three critical times are then:

$$\overline{\kappa}_1 = \frac{12c_1^2 H^4 d^2}{\lambda_+^2} \log\left(\frac{8c_1^2 H^5 d^3}{\lambda_+^2 \delta}\right) \tag{66}$$

$$\overline{\kappa}_2 = \frac{48c_1^2 H^4 d^2}{\lambda_+^2} \log\left(\frac{32c_1^2 H^5 d^3}{\lambda_+^2 \delta}\right) \geq \overline{\kappa}_1 \tag{67}$$

$$\overline{\kappa}_3 = \frac{432c_2^2 H^4 d}{\Delta_{\min}^2 \lambda_+^3} \log\left(\frac{288c_2^2 d^2 H^5}{\Delta_{\min}^2 \lambda_+^2 \delta}\right), \tag{68}$$

and we can take $\overline{\kappa} = \max\{\overline{\kappa}_2, \overline{\kappa}_3\}$. $\qquad\qquad\square$

# E    Representation Selection: Proofs of Section 4

The main ingredient behind the proofs of Theorems 10 and 12 In order to show a regret guarantee for the LSVI-LEADER algorithm, we start by showing a version of Lemma B.4 in [Jin et al., 2020] that takes into account the presence of multiple representations.

First we need the corresponding version of Lemma D.6 in [Jin et al., 2020].

**Lemma 22.** *Given an MDP $M$ and a set of representations $\{\Phi_j\}_{j \in [N]}$ satisfying the low-rank assumption (Asm. 2). Let $\mathcal{V}$ denote a class of functions mapping from $\mathcal{S}$ to $\mathbb{R}$ with the following parametric form,*

$$V(\cdot) = \min\left(\min_{j \in [N]} \max_a \boldsymbol{w}_j^\top \phi_j(\cdot, a) + \beta \sqrt{\phi_j(\cdot, a)^\top \boldsymbol{\Lambda}_j^{-1} \phi_j(\cdot, a)}, H\right)$$

*where the parameters $\{\boldsymbol{w}_j, \boldsymbol{\Lambda}_j\}_{j=1}^N, \beta$ satisfy $\|\boldsymbol{w}\| \leq L$, $\beta \in [0, B]$ and the minimum eigenvalue of $\boldsymbol{\Lambda}_j$ satisfies $\lambda_{\min}(\boldsymbol{\Lambda}_j) \geq \lambda$. Assume $\|\phi(s, a)\| \leq 1$ for all $(s, a)$ pairs and let $\mathcal{N}_\epsilon$ be the $\epsilon$-covering number of $\mathcal{V}$ with respect to the distance $\mathrm{dist}(V, V') = \sup_s |V(s) - V'(s)|$. Then,*

$$\log \mathcal{N}_\epsilon \leq N\left(d \log(1 + 4L/\epsilon) + d^2 \log\left(1 + 8d^{1/2} B^2/(\lambda \epsilon^2)\right)\right)$$

*Proof.* Let's reparametrize the function class $\mathcal{V}$ by $\boldsymbol{A}_j = \beta^2 \boldsymbol{\Lambda}_j^{-1}$, so we have,

$$V(\cdot) = \min\left(\min_{j \in [N]} \max_a \boldsymbol{w}_j^\top \phi_j(\cdot, a) + \sqrt{\phi_j(\cdot, a)^\top \boldsymbol{A}_j \phi_j(\cdot, a)}, H\right) \tag{69}$$

for $\|\boldsymbol{w}_j\| \leq L$ and $\|\boldsymbol{A}_j\| \leq B^2 \lambda^{-1}$. For any two functions $V_1, V_2 \in \mathcal{V}$, let them take the form in Equation 69 with parameters $(\{\boldsymbol{w}_j^{(1)}, \boldsymbol{A}_j^{(1)}\}_{j=1}^N$ and $(\{\boldsymbol{w}_j^{(2)}, \boldsymbol{A}_j^{(2)}\}_{j=1}^N$. Then since $\min_j, \min(\cdot, H)$ and $\max_a$ are contraction maps, we have

$$\mathrm{dist}(V_1, V_2) \leq \sup_{j,s,a} \left| \left[\left(\boldsymbol{w}_j^{(1)}\right)^\top \phi_j(\cdot, a) + \sqrt{\phi_j(\cdot, a)^\top \boldsymbol{A}_j^{(1)} \phi_j(\cdot, a)}\right] - \right. \tag{70}$$

$$\left. \left[\left(\boldsymbol{w}_j^{(2)}\right)^\top \phi_j(\cdot, a) + \sqrt{\phi_j(\cdot, a)^\top \boldsymbol{A}_j^{(2)} \phi_j(\cdot, a)}\right] \right|$$

$$\leq \sup_j \left(\sup_{\|\phi_j\| \leq 1} \left| \left[\left(\boldsymbol{w}_j^{(1)}\right)^\top \phi_j + \sqrt{\phi_j^\top \boldsymbol{A}_j^{(1)} \phi_j}\right] - \left[\left(\boldsymbol{w}_j^{(2)}\right)^\top \phi_j + \sqrt{\phi_j^\top \boldsymbol{A}_j^{(2)} \phi_j}\right] \right| \right)$$

$$\leq \sup_j \left(\sup_{\|\phi_j\| \leq 1} \left| \left(\boldsymbol{w}_j^{(1)} - \boldsymbol{w}_j^{(2)}\right)^\top \phi_j \right| + \sup_{\|\phi_j\| \leq 1} \sqrt{\left|\phi_j^\top \left(\boldsymbol{A}_j^{(1)} - \boldsymbol{A}_j^{(2)}\right) \phi_j\right|} \right)$$

$$= \sup_j \|\boldsymbol{w}_j^{(1)} - \boldsymbol{w}_j^{(2)}\| + \sqrt{\|\boldsymbol{A}_j^{(1)} - \boldsymbol{A}_j^{(2)}\|}$$

$$\leq \sup_j \|\boldsymbol{w}_j^{(1)} - \boldsymbol{w}_j^{(2)}\| + \sqrt{\|\boldsymbol{A}_j^{(1)} - \boldsymbol{A}_j^{(2)}\|_F} \tag{71}$$

For matrices $\|\cdot\|$ and $\|\cdot\|_F$ denote the matrix operator norm and the frobenius norm respectively.

Let $\mathcal{C}_j^{\boldsymbol{w}}$ be an $\epsilon/2$ cover of $\{\boldsymbol{w}_j \in \mathbb{R}^d | \|\boldsymbol{w}_j\| \leq L\}$ with respect to the 2-norm and let $\mathcal{C}_j^{\boldsymbol{A}}$ be an $\epsilon^2/4-$cover of $\{\boldsymbol{A} \in \mathbb{R}^{d\times d} | \|\boldsymbol{A}\|_F \leq d^{1/2}B^2\lambda^{-1}\}$ with respect to the Frobenius norm. By Lemma D.5. in [Jin et al., 2020] we know that,

$$|\mathcal{C}_j^{\boldsymbol{w}}| \leq (1 + 4L/\epsilon)^d, \qquad |\mathcal{C}_j^{\boldsymbol{A}}| \leq \left(1 + 8d^{1/2}B^2/(\lambda\epsilon^2)\right)^{d^2}$$

By Equation 71, for any $V_1 \in \mathcal{V}$ there exists points $\{\boldsymbol{w}_j^{(2)}\}_{j=1}^N$ and $\{\boldsymbol{A}_j^{(2)}\}_{j=1}^N$ such that $V_2$ parametrized by $(\{\boldsymbol{w}_j^{(2)}\}_{j=1}^N, \boldsymbol{A}_j^{(2)}\}_{j=1}^N)$ satisfies $\mathrm{dist}(V_1, V_2) \leq \epsilon$. Hence it holds that $\mathcal{N}_\epsilon \leq \left(|\mathcal{C}_j^{\boldsymbol{w}}||\mathcal{C}_j^{\boldsymbol{A}}|\right)^N$, which gives:

$$\log \mathcal{N}_\epsilon \leq N\left(d\log(1 + 4L/\epsilon) + d^2\log\left(1 + 8d^{1/2}B^2/(\lambda\epsilon^2)\right)\right).$$

$\square$

**Lemma 23** (Multi-representation version of Lemma B.3 in [Jin et al., 2020]). *Given an MDP $M$ and a set of representations $\{\Phi_j\}_{j\in[N]}$ satisfying the low-rank assumption (Asm. 2). For all $k \in \mathbb{N}, h \in [H]$, with probability $1 - 2\delta$:*

$$\left\|\sum_{i=1}^k \phi_h^{(j)}(s_h^i, a_h^i)\left(\overline{V}_{h+1}^k(s_{h+1}^i) - \mathbb{P}_h\overline{V}_{h+1}^k(s_h^i, a_h^i)\right)\right\|_{\Lambda_{h,k}^{-1}(j)} \leq CdH\sqrt{N\log(2N(c_\beta + 1)dHk/\delta)}, \tag{72}$$

*for all $j \in [N]$ and for some constant $C$ independent of $c_\beta$.*

*Proof.* This result follows from a simple use of an anytime version of Lemma D.4 from [Jin et al., 2020] with $\epsilon = dH/k$ and $\delta' = \frac{\delta}{2N}$ and $\lambda = 1$. Let $j \in [N]$ be one of the representations.

$$\left\|\sum_{i=1}^k \phi_h^{(j)}(s_h^i, a_h^i)\left(\overline{V}_{h+1}^k(s_{h+1}^i) - \mathbb{P}_h\overline{V}_{h+1}^k(s_h^i, a_h^i)\right)\right\|_{\Lambda_{h,k}^{-1}(j)}^2$$
$$\leq 4H^2\left[\frac{d}{2}\log\left(\frac{k+\lambda}{\lambda}\right) + 2\log\frac{\pi k}{\sqrt{6}} + \log\frac{2}{\delta} + dN\log\left(1 + \frac{8k^{3/2}}{\sqrt{\lambda d}}\right) + \right.$$
$$\left. d^2N\log\left(1 + \frac{8\sqrt{d}c_\beta^2 k^2\log(2dHk/\delta)}{\lambda}\right)\right] + \frac{8d^2H^2}{\lambda}$$
$$= \mathcal{O}(d^2NH^2\log(2N(c_\beta + 1)dHk/\delta))$$

A simple union bound over all representations in $\{\Phi_j\}_{j\in[N]}$ yields the desired result.

$\square$

We have now the necessary ingredients to prove an equivalent version to Lemma B.4 from [Jin et al., 2020] for the case of multiple representations.

**Lemma 24** (Equivalent to Lemma B.4 in [Jin et al., 2020]). *Given an MDP $M$ and a set of representations $\{\Phi_j\}_{j\in[N]}$ satisfying the low-rank assumption (Asm. 2). With probability at least $1 - 2\delta$, for any policy $\pi$, any episode $k \in \mathbb{N}$, stage $h \in [H]$, state $s \in \mathcal{S}$ and action $a \in \mathcal{A}$,*

$$\left|\langle\phi_h^{(j)}(s,a), \boldsymbol{w}_h^k(j)\rangle - Q_h^\pi(s,a) - \mathbb{P}_h\left(\overline{V}_{h+1}^k - V_{h+1}^\pi\right)(s,a)\right| \leq \beta_k\left\|\phi^{(j)}(s,a)\right\|_{\Lambda_{h,k}(j)^{-1}}$$

*where $\beta_k = C'dH\sqrt{N\log(2N(c_\beta + 1)dHk/\delta)}$. For some absolute constant $C'$.*

*Proof.* We know that for any $(s, a, h) \in \mathcal{S} \times \mathcal{A} \times [H]$:

$$Q_h^\pi(s, a) = \langle \phi_h^{(j)}(s, a), \boldsymbol{w}_h^\pi(j) \rangle = \left( r_h + \mathbb{P}_h V_{h+1}^\pi \right)(s, a) \quad \forall j \in [N],$$

This gives

$$\boldsymbol{w}_h^k(j) - \boldsymbol{w}_h^\pi(j) = \Lambda_{h,k}^{-1}(j) \sum_{i=1}^{k-1} \phi_h^{(j)}(s_h^i, a_h^i) \left( r_h(s_h^i, a_h^i) + \max_{a \in \mathcal{A}} \overline{Q}_{h+1}^{k-1}(s_{h+1}^i, a) \right) - \boldsymbol{w}_h^\pi$$

$$= \Lambda_{h,k}(j)^{-1} \left\{ -\lambda \boldsymbol{w}_h^\pi + \sum_{i=1}^{k-1} \phi_h^{(j)}(s_h^i, a_h^i) \left( \overline{V}_{h+1}^k(s_{h+1}^i) - \mathbb{P}_h V_{h+1}^\pi(s_h^i, a_h^i) \right) \right\}$$

$$= \underbrace{-\lambda \Lambda_{h,k}^{-1}(j) \boldsymbol{w}_h^\pi(j)}_{\boldsymbol{q}_1} + \underbrace{\Lambda_{h,k}^{-1}(j) \sum_{i=1}^{k-1} \phi_h^{(j)}(s_h^i, a_h^i) \left( \overline{V}_{h+1}^k(s_{h+1}^i) - \mathbb{P}_h \overline{V}_{h+1}^k(s_h^i, a_h^i) \right)}_{\boldsymbol{q}_2} +$$

$$\underbrace{\Lambda_{h,k}^{-1}(j) \left( \sum_{i=1}^{k-1} \phi_h^{(j)}(s_h^i, a_h^i) \mathbb{P}_h \left( \overline{V}_{h+1}^k - V_{h+1}^\pi \right)(s_h^i, a_h^i) \right)}_{\boldsymbol{q}_3}$$

Now we bound the terms on the right hand side. For the first term,

$$\left| \langle \phi_h^{(j)}(s, a), \boldsymbol{q}_1 \rangle \right| = \left| \lambda \langle \phi_h^{(j)}(s, a), \Lambda_{h,k}^{-1}(j) \boldsymbol{w}_h^\pi \rangle \right| \leq \sqrt{\lambda} \| \boldsymbol{w}_h^\pi \| \left\| \phi_h^{(j)}(s, a) \right\|_{\Lambda_{h,k}^{-1}(j)} \overset{(i)}{\leq} 2H\sqrt{d\lambda} \left\| \phi_h^{(j)}(s, a) \right\|_{\Lambda_{h,k}^{-1}(j)}$$

Inequality $(i)$ above holds because of Lemma B.1 of [Jin et al., 2020]. For the second term $\boldsymbol{q}_2$, given the event defined in Lemma 23 (which holds with probability at least $1 - 2\delta$) we have,

$$\left| \langle \phi_h^{(j)}(s, a), \boldsymbol{q}_2 \rangle \right| \leq CdH \sqrt{N \log(2N(c_\beta + 1)dHk/\delta)} \left\| \phi_h^{(j)}(s, a) \right\|_{\Lambda_{h,k}^{-1}(j)}$$

For the third term,

$$\langle \phi_h^{(j)}(s, a), \boldsymbol{q}_3 \rangle$$

$$= \left\langle \phi_h^{(j)}(s, a), \left( \Lambda_{h,k}^{-1}(j) \right) \sum_{i=1}^{k-1} \phi_h^{(j)}(s_h^i, a_h^i) \mathbb{P}_h \left( \overline{V}_{h+1}^k - V_{h+1}^\pi \right)(s_h^i, a_h^i) \right\rangle$$

$$= \left\langle \phi_h^{(j)}(s, a), \left( \Lambda_{h,k}^{-1}(j) \right) \sum_{i=1}^{k-1} \phi_h^{(j)}(s_h^i, a_h^i) \phi_j^\top(s_h^i, a_h^i) \int \left( \overline{V}_{h+1}^k - V_{h+1}^\pi \right)(s_{h+1}') d\boldsymbol{\mu}_h^j(s_{h+1}' | s_h^i, a_h^i) \right\rangle$$

$$= \underbrace{\left\langle \phi_h^{(j)}(s, a), \int \left( \overline{V}_{h+1}^k - V_{h+1}^\pi \right)(s_{h+1}') d\boldsymbol{\mu}_h^j(s_{h+1}' | s_h^i, a_h^i) \right\rangle}_{p_1} -$$

$$\underbrace{\lambda \left\langle \phi_h^{(j)}(s, a), \Lambda_{h,k}^{-1}(j) \int \left( \overline{V}_{h+1}^k - V_{h+1}^\pi \right)(s_{h+1}') d\boldsymbol{\mu}_h^j(s_{h+1}' | s_h^i, a_h^i) \right\rangle}_{p_2}$$

And therefore,

$$p_1 = \mathbb{P}_h \left( \overline{V}_{h+1}^k - V_{h+1}^\pi \right)(s, a), \qquad |p_2| \leq 2H\sqrt{d\lambda} \left\| \phi_h^{(j)}(s, a) \right\|_{\Lambda_{h,k}^{-1}(j)}$$

Finally since $\langle \phi_h^{(j)}(s,a), \boldsymbol{w}_h^k(j) \rangle - Q_h^\pi(s,a) = \langle \phi_h^{(j)}(s,a), \boldsymbol{w}_h^k - \boldsymbol{w}_h^\pi \rangle = \langle \phi_h^{(j)}(s,a), \boldsymbol{q}_1 + \boldsymbol{q}_2 + \boldsymbol{q}_3 \rangle$, we have

$$\left| \langle \phi_h^{(j)}(s,a), \boldsymbol{w}_h^k(j) \rangle - Q_h^\pi(s,a) - \mathbb{P}_h \left( \overline{V}_{h+1}^k - V_{h+1}^\pi \right)(s,a) \right|$$

$$\leq \left( CdH\sqrt{N\log(2N(c_\beta+1)dHk/\delta)} + 4H\sqrt{d\lambda} \right) \left\| \phi_h^{(j)}(s,a) \right\|_{\Lambda_{h,k}^{-1}(j)}$$

$$\leq C'dH\sqrt{N\log(2N(c_\beta+1)dHk/\delta)} \left\| \phi_h^{(j)}(s,a) \right\|_{\Lambda_{h,k}^{-1}(j)}$$

For some constant $C'$. The result follows.

$\square$

**Lemma 25.** *Given an MDP $M$ and a set of representations $\{\Phi_j\}_{j\in[N]}$ satisfying the low-rank assumption (Asm. 2). With probability at least $1 - 2\delta$, for any episode $k \in \mathbb{N}$, stage $h \in [H]$, and state $s \in \mathcal{S}$,*

$$\overline{V}_h^k(s) - V_h^{\pi^k}(s) \leq 2\beta_k \min_{j\in[N]} \|\phi_h^{(j)}(s,\pi_h^k(s))\|_{\Lambda_{h,k}^{-1}(j)} + \mathbb{E}_{s'\sim p_h(s,\pi_h^k(s))} \left[ \overline{V}_{h+1}^k(s') - V_{h+1}^{\pi^k}(s') \right].$$

*Where $\beta_k = C'dH\sqrt{N\log(2N(c_\beta+1)dHk/\delta)}$.*

*Proof.* Note that $\overline{V}_h^k(s) - V_h^{\pi^k}(s) = \overline{Q}_h^k(s,\pi_h^k(s)) - Q_h^{\pi^k}(s,\pi_h^k(s))$. Using Lemma 24, for any $j \in [N]$

$$Q_h^{\pi^k}(s,\pi_h^k(s)) \geq \langle \phi_h^{(j)}(s,\pi_h^k(s)), \boldsymbol{w}_h^k(j) \rangle -$$

$$\mathbb{E}_{s'\sim p_h(s,\pi_h^k(s))}[\overline{V}_{h+1}^k(s') - V_{h+1}^{\pi^k}(s')] - \beta_{h,k}\|\phi_h^{(j)}(s,\pi_h^k(s))\|_{\Lambda_{h,k}^{-1}(j)}$$

And therefore for all $j \in [N]$,

$$\langle \phi_h^{(j)}(s,\pi_h^k(s)), \boldsymbol{w}_h^k(j) \rangle + \beta_{h,k}\|\phi_h^{(j)}(s,\pi_h^k(s))\|_{\Lambda_{h,k}^{-1}(j)} - V_h^{\pi^k}(s) \leq$$

$$2\beta_{h,k}\|\phi_h^{(j)}(s,\pi_h^k(s))\|_{\Lambda_{h,k}^{-1}(j)} + \mathbb{E}_{s'\sim p_h(s,\pi_h^k(s))}[\overline{V}_{h+1}^k(s') - V_{h+1}^{\pi^k}(s')]$$

Taking the minimum over $j \in [N]$ (and $H$) on the LHS yields the result,

$$\overline{V}_h^k(s) - V_h^{\pi^k}(s) \leq 2\beta_{h,k} \min_{j\in[N]} \|\phi_h^{(j)}(s,\pi_h^k(s))\|_{\Lambda_{h,k}^{-1}(j)} + \mathbb{E}_{s'\sim p_h(s,\pi_h^k(s))}[\overline{V}_{h+1}^k(s') - V_{h+1}^{\pi^k}(s')].$$

$\square$

Finally we show this implies optimism holds,

**Lemma 26.** *[Optimism. Equivalent version of Lemma B.5 in [Jin et al., 2020]] With probability $1 - \delta$ and for all $s, a \in \mathcal{S} \times \mathcal{A}$, $k \in \mathbb{N}$ and $h \in [H]$, the $\{\overline{Q}_h^k\}_{h\in[H]}$ functions of LSVI-LEADER satisfy,*

$$\overline{Q}_h^k(s,a) \geq Q_h^*(s,a).$$

*Proof.* The same proof as in Lemma B.5 in [Jin et al., 2020] works just simply modifying it to have a minimum over $j \in [N]$ in the necessary places. We reproduce the argument here for completeness. The proof of the Lemma proceeds by induction.

First, we prove the base case, at the last step $H$. The statement holds because $\overline{Q}_H^k(s,a) \geq Q_H^*(s,a)$ since the value function at $H + 1$ is zero and by Lemma 24 we have that with probability at least $1 - 2\delta$ for all $k \in \mathbb{N}$, $s \in \mathcal{S}, a \in \mathcal{A}$ and any $j \in [N]$,

$$\left| \langle \phi_h^{(j)}(s,a), \boldsymbol{w}_H^k(j) \rangle - Q_H^{\pi^*}(s,a) \right| \leq C'dH\sqrt{N\log(2N(c_\beta+1)dHk/\delta)} \left\| \phi_h^{(j)}(s,a) \right\|_{\Lambda_{H,k}^{-1}(j)}$$

Therefore for all $j \in [N]$, with probability at least $1 - 2\delta$,

$$\langle \phi_h^{(j)}(s,a), \boldsymbol{w}_H^k(j) \rangle + C' dH \sqrt{N \log(2N(c_\beta + 1)dHk/\delta)} \left\| \phi_h^{(j)}(s,a) \right\|_{\Lambda_{H,k}^{-1}(j)} \geq Q_H^{\pi_*}(s,a)$$

Since $H \geq Q_H^{\pi_*}(s,a)$ by definition, we conclude that taking the mimimum over $j \in [N]$ (and $H$), and using the fact that

$$\overline{Q}_h^k(s,a) = \min \left( \min_{j \in [N]} \langle \phi_h^{(j)}(s,a), \boldsymbol{w}_H^k(j) \rangle + C' dH \sqrt{N \log(2N(c_\beta + 1)dHk/\delta)} \left\| \phi_h^{(j)}(s,a) \right\|_{\Lambda_{H,k}^{-1}(j)}, H \right)$$

We conclude that,

$$\overline{Q}_H^k(s,a) \geq Q_H^{\pi_*}(s,a).$$

Now, suppose the statement holds true at step $h + 1$ and consider step $h$. Again by Lemma 24 we have, for all $k \in [K]$ and all $j \in [N]$

$$\left| \langle \phi_h^{(j)}(s,a), \boldsymbol{w}_h^k(j) \rangle - Q_h^{\pi_*}(s,a) - \mathbb{P}_h \left( \overline{V}_{h+1}^k - V_{h+1}^{\pi_*} \right)(s,a) \right|$$

$$\leq C' dH \sqrt{N \log(2N(c_\beta + 1)dHk/\delta)} \left\| \phi_j(s,a) \right\|_{\Lambda_{h,k}^{-1}(j)}$$

By the induction assumption that $\mathbb{P}_h \left( \overline{V}_{h+1}^k - V_{h+1}^{\pi_*} \right)(s,a) \geq 0$, we have for all $j \in [N]$:

$$Q_h^{\pi_*}(s,a) \leq \min \left( \langle \phi_h^{(j)}(s,a), \boldsymbol{w}_h^k(j) \rangle + C' dH \sqrt{N \log(2N(c_\beta + 1)dHk/\delta)} \left\| \phi_h^{(j)}(s,a) \right\|_{\Lambda_{h,k}^{-1}(j)}, H \right)$$

The result follows by taking a minimum over $j \in [N]$. $\qquad \square$

**Finishing the proof of Theorem 10.** Having proven Lemma 25 and that optimism holds for LSVI-LEADER (Lemma 26), we conclude that an equivalent version of Assumption 16 holds. The same logic of the proofs of Lemmas 17, 18 and Theorem 19 apply in this case. Hence, we conclude that the regret of LSVI-LEADER is upper bounded by the minimum of these regret bounds for all representations $z \in \mathcal{Z}$, thus proving the first result. To obtain the second result, simply notice that, if $z^\star \in \mathcal{Z}$ is UNISOFT, then we can use the refined analysis for LSVI-UCB of Thm. 9 to show that $\widetilde{R}(K, z^\star, \{\beta_k\})$ is upper bounded by a constant independent of $K$, hence proving constant regret for LSVI-LEADER.

**Proof of Theorem 12.** The proof follows the template of Thm 9, but as shown in Lemma 25, the confidence sets of LSVI-LEADER scale with the minimum w.r.t. $j$ of the feature norms. In place of Equation 3, and with the aid of Lemma 33 we see that since the collection of feature maps $\{\Phi_j\}_{j \in [M]}$ is UNISOFT-mixing for all reachable $s, a$:

$$\beta_k \min_{j \in [N]} \left\| \phi_h^{(j)}(s,a) \right\|_{\Lambda_{h,k}^{-1}(j)} \leq \beta_k \frac{k + \lambda - g(k) - 8\sqrt{k \log(2NdHk/\delta)}}{(k\lambda^+(h,s,a) + \lambda - g(k) - 8\sqrt{k \log(2NdHk/\delta)})^{3/2}} \tag{73}$$
$$= \widetilde{O}(k^{-1/2}),$$

where $g(k) = \widetilde{O}(\sqrt{k})$ is the regret upper bound from Thm. 10,

$$\lambda^+(h,s,a) = \max_{j \in \mathcal{J}(h,s,a)} \lambda_{h,j}^+, \tag{74}$$

and $\mathcal{J}(h,s,a) \subseteq [N]$ is such that $j \in \mathcal{J}(h,s,a)$ if $\phi_h^{(j)}(s,a) \in \text{span}\left\{ \phi_h^{(j)}(s, \pi_h^*(s)) | \rho_h^\star(s) > 0 \right\}$. To see this, notice that we can instantiate Lemma 33 with any representation $j \in [N]$ such that $\phi_h^{(j)}(s,a)$ belongs to the span of optimal features. So we use the representation with the largest eigenvalue $\lambda_{h,j}^+$. The UNISOFT-mixing property (Def. 11) guarantees $\mathcal{J}(h,s,a)$ is always nonempty.

By (73) and Lemma 18 (where $C_h = 0$ thanks to local optimism), for each $h \in [H]$ there exists an episode $\kappa_h$ independent of $K$ such that, for all reachable $s$ and $k > \kappa_h$:

$$\Delta_h(s, \pi_h^k(s)) \leq 2\beta_k \mathbb{E}_{\pi^k} \left[ \sum_{i=h}^{H} \frac{k + \lambda - g(k) - 8\sqrt{k \log(2NdHk/\delta)}}{(k\lambda^+(i, s_i, a_i) + \lambda - g(k) - 8\sqrt{k \log(2NdHk/\delta)})^{3/2}} \,\middle|\, s_h = s \right]$$
$$< \Delta_{\min}. \tag{75}$$

So after $\widetilde{\kappa} = \max_h \{\kappa_h\}$ episodes, LSVI-UCB suffers zero regret. Finally, the regret up to $\widetilde{\kappa}$ cannot be worse than that obtained in Thm. 10 without the UNISOFT-mixing property.

# F  Auxiliary Results

**Proposition 27** (Azuma's inequality). *Let $\{(Z_t, \mathcal{F}_t)\}_{t \in \mathbb{N}}$ be a martingale difference sequence such that $|Z_t| \leq a$ almost surely for all $t \in \mathbb{N}$. Then, for all $\delta \in (0, 1)$,*

$$\mathbb{P}\left(\forall t \geq 1 : \left|\sum_{k=1}^{t} Z_k\right| \leq a\sqrt{t \log(2t/\delta)}\right) \geq 1 - \delta. \tag{76}$$

**Proposition 28** (Matrix Azuma, Tropp, 2012). *Let $\{X_k\}_{k=1}^{t}$ be a finite adapted sequence of symmetric matrices of dimension $d$, and $\{C_k\}_{k=1}^{t}$ a sequence of symmetric matrices such that for all $k$, $\mathbb{E}_k[X_k] = 0$ and $X_k^2 \preceq C_k^2$ almost surely. Then, with probability at least $1 - \delta$:*

$$\lambda_{\max}\left(\sum_{k=1}^{t} X_k\right) \leq \sqrt{8\sigma^2 \log(d/\delta)}, \tag{77}$$

*where $\sigma^2 = \left\|\sum_{k=1}^{t} C_k^2\right\|$.*

**Proposition 29** (He et al. [2020]). *For any $h \in [H]$, $s \in \mathcal{S}$, and $\pi \in \Pi$:*

$$V_h^\star(s) - V_h^\pi(s) = \mathbb{E}_\pi\left[\sum_{i=h}^{H} \Delta_i(s_i, a_i)\middle| s_h = s\right],$$

*Hence the regret after $K$ episodes can be expressed as:*

$$R(K) = \sum_{k=1}^{K} V_1^\star(s_1^k) - V_1^{\pi^k}(s_1^k) = \sum_{k=1}^{K} \mathbb{E}_{\pi^k}\left[\sum_{h=1}^{H} \Delta_h(s_h, a_h)\middle| s_1 = s_1^k\right].$$

*Proof.* By definition of $\Delta_h$:

$$V_h^\star(s) - V_h^\pi(s) = Q_h^\star(s, \pi_h(s)) + \Delta_h(s, \pi_h(s)) - V_h^\pi(s) \tag{78}$$
$$= r_h(s, \pi_h(s)) + \mathbb{E}_{s' \sim p_h(s, \pi_h(s))}[V_{h+1}^\star(s')] + \Delta_h(s, \pi_h(s)) - r_h(s, \pi_h(s))$$
$$- \mathbb{E}_{s' \sim \mathbb{P}_h(s_h, \pi_h(s_h))}[V_{h+1}^\pi(s')] \tag{79}$$
$$= \Delta_h(s_h, \pi_h(s_h)) + \mathbb{E}_{s' \sim \mathbb{P}_h(s_h, \pi_h(s_h))}[V_{h+1}^\star(s') - V_{h+1}^\pi(s')]. \tag{80}$$

Unrolling the recursion up to $H$ concludes the proof. $\qquad\square$

**Lemma 30.** *Assume $R(k) \leq g(k)$ for all $k \geq 1$ and Asm. 3 holds. Then, probability $1 - \delta$, for all $h, k$:*

$$\Lambda_h^{k+1} \succeq k\Lambda_h^\star + \lambda I - \Delta_{\min}^{-1} g(k) I - 8L^2 I \sqrt{k \log(2dkH/\delta)}. \tag{81}$$

*Proof.* Define a trajectory as a sequence of states and actions $\tau_h = (s_1, a_1, \ldots, s_h, a_h)$. Let $\Gamma_h$ denote the set of all trajectories of length $h$. The distribution over trajectories induced by a (deterministic) policy $\pi$ is $p_h^\pi(\tau_h) = \mu(s_1)\mathbb{1}\{a_1 = \pi_1(s_1)\} p_1(s_2|s_1, a_1) \ldots p_{h-1}(s_h|s_{h-1}, a_{h-1})\mathbb{1}\{a_h = \pi_h(s_h)\}$. We abbreviate as $p_h^\star$ the distribution induced by the optimal policy $\pi^\star$ and as $p_h^k$ the one induced by $\pi^k$, the algorithm's policy at episode $k$. Let us define the following event:

$$E_h^k = \{\tau \in \Gamma_h \text{ s.t. } a_i = \pi_h^k(s_i) = \pi_h^\star(s_i) \text{ for } i = 1, \ldots, h\}. \tag{82}$$

Then:

$$
\begin{aligned}
\Lambda_h^{k+1} - \lambda I &= \sum_{i=1}^k \phi(s_h^i, a_h^i)\phi(s_h^i, a_h^i)^\mathsf{T} \\
&\succeq \sum_{i=1}^k \mathbb{1}\left\{\tau_h^i \in E_h^i\right\} \phi(s_h^i, a_h^i)\phi(s_h^i, a_h^i)^\mathsf{T} \\
&= \sum_{i=1}^k \mathbb{1}\left\{\tau_h^i \in E_h^i\right\} \phi_h^\star(s_h^i)\phi_h^\star(s_h^i)^\mathsf{T} \qquad (83) \\
&= \underbrace{\sum_{i=1}^k \mathbb{E}_{\tau_h \sim p_h^i}\left[\mathbb{1}\left\{\tau_h \in E_h^i\right\} \phi_h^\star(s_h)\phi_h^\star(s_h)^\mathsf{T}\right]}_{(A)} \\
&\quad + \underbrace{\sum_{i=1}^k \left(\mathbb{1}\left\{\tau_h^i \in E_h^i\right\} \phi_h^\star(s_h^i)\phi_h^\star(s_h^i)^\mathsf{T} - \mathbb{E}_{\tau_h \sim p_h^i}\left[\mathbb{1}\left\{\tau_h \in E_h^i\right\} \phi_h^\star(s_h)\phi_h^\star(s_h)^\mathsf{T}\right]\right)}_{(B)},
\end{aligned}
$$

where (83) is by definition of $E_h^i$ and expectations are conditioned on history up to the beginning of the $i$-th episode. We first bound $(B)$ with a matrix version of Azuma's inequality. Let:

$$
X_h^i = \mathbb{1}\left\{\tau_h^i \in E_h^i\right\} \phi_h^\star(s_h^i)\phi_h^\star(s_h^i)^\mathsf{T} - \mathbb{E}_{\tau_h \sim p_h^i}\left[\mathbb{1}\left\{\tau_h \in E_h^i\right\} \phi_h^\star(s_h)\phi_h^\star(s_h)^\mathsf{T}\right].
$$

Clearly $\mathbb{E}[X_h^i] = 0$. Moreover, since $X_h^i$ is symmetric:

$$
(X_h^i)^2 \preceq \lambda_{\max}((X_h^i)^2)I \preceq \left\|X_h^i\right\|^2 I \preceq 4I. \qquad (84)
$$

Then by Proposition 28, with probability $1 - \delta_h^k$:

$$
\lambda_{\max}\left(\sum_{i=1}^k X_h^i\right) \leq 4\sqrt{2k\log(d/\delta_h^k)}. \qquad (85)
$$

Setting $\delta_h^k = \delta/(2Hk^2)$ we can perform a union bound over episodes and stages to obtain, with probability $1 - \delta$, for all $h, k$:

$$
(B) = \sum_{i=1}^k X_h^i \preceq \lambda_{\max}\left(\sum_{i=1}^k X_h^i\right) I \preceq 8I\sqrt{k\log(2dHk/\delta)}. \qquad (86)
$$

Now we focus on the $(A)$ term. First, observe that the probability measures $p_h^k$ and $p_h^\star$ agree on $E_h^k$. Indeed, if $\tau_h \in E_h^k$:

$$
\begin{aligned}
p_h^k(\tau_h) &= \mu(s_1)\mathbb{1}\left\{a_1 = \pi_1^k(s_1)\right\} p_1(s_2|s_1, a_1)\ldots p_{h-1}(s_h|s_{h-1}, a_{h-1})\mathbb{1}\left\{a_h = \pi_h^k(s_h)\right\} \\
&= \mu(s_1)\mathbb{1}\left\{a_1 = \pi_1^\star(s_1)\right\} p_1(s_2|s_1, a_1)\ldots p_{h-1}(s_h|s_{h-1}, a_{h-1})\mathbb{1}\left\{a_h = \pi_h^\star(s_h)\right\} \qquad (87) \\
&= \mu(s_1)p_1(s_2|s_1, a_1)\ldots p_{h-1}(s_h|s_{h-1}, a_{h-1}). \qquad (88)
\end{aligned}
$$

So:

$$(A) = \sum_{i=1}^{k} \mathbb{E}_{\tau_h \sim p_h^i} [\mathbb{1}\left\{\tau_h \in E_h^i\right\} \phi_h^\star(s_h)\phi_h^\star(s_h)^\mathsf{T}]$$

$$= \sum_{i=1}^{k} \mathbb{E}_{\tau_h \sim p_h^\star} [\mathbb{1}\left\{\tau_h \in E_h^i\right\} \phi_h^\star(s_h)\phi_h^\star(s_h)^\mathsf{T}] \tag{89}$$

$$= k \, \mathbb{E}_{\tau_h \sim p_h^\star} [\phi_h^\star(s_h)\phi_h^\star(s_h)^\mathsf{T}] - \sum_{i=1}^{k} \int_{\Gamma_h \setminus E_h^i} \phi_h^\star(s_h)\phi_h^\star(s_h)^\mathsf{T} p_h^\star(\mathrm{d}\tau_h) \tag{90}$$

$$= k \, \mathbb{E}_{s \sim \rho_h^\star} [\phi_h^\star(s_h)\phi_h^\star(s_h)^\mathsf{T}] - \sum_{i=1}^{k} \int_{\Gamma_h \setminus E_h^i} \phi_h^\star(s_h)\phi_h^\star(s_h)^\mathsf{T} p_h^\star(\mathrm{d}\tau_h) \tag{91}$$

$$\succeq k \, \mathbb{E}_{s \sim \rho_h^\star} [\phi_h^\star(s_h)\phi_h^\star(s_h)^\mathsf{T}] - I \sum_{i=1}^{k} \left(1 - \int_{E_h^i} p_h^\star(\mathrm{d}\tau_h)\right) \tag{92}$$

$$= k \, \mathbb{E}_{s \sim \rho_h^\star} [\phi_h^\star(s_h)\phi_h^\star(s_h)^\mathsf{T}] - I \sum_{i=1}^{k} \left(1 - \int_{E_h^i} p_h^\star(\mathrm{d}\tau_h)\right) \tag{93}$$

$$= k \, \mathbb{E}_{s \sim \rho_h^\star} [\phi_h^\star(s_h)\phi_h^\star(s_h)^\mathsf{T}] - I \underbrace{\sum_{i=1}^{k} \mathbb{E}_{\tau_h \sim p_h^i(\tau_h)} [\mathbb{1}\left\{\tau_h \notin E_h^i\right\}]}_{(C)}. \tag{94}$$

Finally, under Asm. 3 and the regret upper bound:

$$(C) = \sum_{i=1}^{k} \mathbb{E}_{\tau_h \sim p_h^i(\tau_h)} [\mathbb{1}\left\{\tau_h \notin E_h^i\right\}]$$

$$\leq \sum_{i=1}^{k} \sum_{j=1}^{h} \mathbb{E}_{\pi^i} [\mathbb{1}\left\{a_j \neq \pi_j^\star(s_j)\right\}] \tag{95}$$

$$\leq \sum_{i=1}^{k} \sum_{j=1}^{h} \mathbb{E}_{\pi^i} [\mathbb{1}\left\{\Delta_j(s_j, a_j) \geq \Delta\right\}] \tag{96}$$

$$\leq \sum_{i=1}^{k} \sum_{j=1}^{h} \mathbb{E}_{\pi^i} \left[\frac{\Delta_j(s_j, a_j)}{\Delta_{\min}}\right] \tag{97}$$

$$= \frac{1}{\Delta_{\min}} \sum_{i=1}^{k} \mathbb{E}_{\pi^i} \left[\sum_{j=1}^{h} \Delta_j(s_j, a_j)\right] \tag{98}$$

$$\leq \frac{1}{\Delta_{\min}} \sum_{i=1}^{k} \mathbb{E}_{\pi^i} \left[\sum_{h=1}^{H} \Delta_h(s_h, a_h)\right] \tag{99}$$

$$= \frac{R(k)}{\Delta_{\min}} \leq \frac{g(k)}{\Delta_{\min}}, \tag{100}$$

where (95) is by definition of $E_h^i$, (96) is from the uniqueness of the optimal policy and Asm. 3, and (100) is from Proposition 29. $\qquad\square$

**Proposition 31** (Lemma 29 from [Papini et al., 2021]). *Let $v \in \mathbb{R}^d$ with $\|v\| = 1$ and $A \in \mathbb{R}^{d \times d}$ symmetric invertible with non-zero eigenvalues $\lambda_1 \leq \cdots \leq \lambda_d$ and corresponding orthonormal eigenvectors $u_1, \ldots, u_d$. Let $\mathcal{I} \subseteq [d]$ be any index set. If $v \in \mathrm{span}\{u_i\}_{i \in \mathcal{I}}$ and $\lambda_i > 0$ for all $i \in \mathcal{I}$:*

$$v^\mathsf{T} A^{-1} v \leq \frac{(\max_{i \in \mathcal{I}} \lambda_i + \min_{i \in \mathcal{I}} \lambda_i)^2}{4 \max_{i \in \mathcal{I}} \lambda_i \min_{i \in \mathcal{I}} \lambda_i} \frac{1}{v^\mathsf{T} A v}.$$

**Proposition 32** (e.g., Lemma 30 from [Papini et al., 2021]). *The smallest nonzero eigenvalue of a symmetric p.s.d. matrix $A \in \mathbb{R}^{d \times d}$ is:*

$$\lambda_{\min}^+(A) = \min_{\substack{\boldsymbol{v} \in \mathrm{Im}(A) \\ \|\boldsymbol{v}\| = 1}} \boldsymbol{v}^\mathsf{T} A \boldsymbol{v},$$

*where $\mathrm{Im}(A)$ denotes the column space of $A$.*

**Lemma 33.** *Consider a d-dimensional representation $(\phi_h)_{h \in [H]}$. Assume there exists an increasing $\widetilde{O}(\sqrt{k})$ function $g$ such that $R(k) \leq g(k)$ for all $k \geq 1$, Asm. 3 holds, and $\beta_k = \widetilde{O}(1)$. Then with probability $1 - \delta$, for all $h$, there exists a constant $\widetilde{\kappa}_h$ such that, for every $k \geq \widetilde{\kappa}_h$ and all $s, a$ having $\phi_h(s, a) \in \mathrm{span}\{\phi_h^\star(s) | \rho_h^\star(s) > 0\}$,*

$$\beta_k \|\phi_h(s,a)\|_{(\Lambda_h^k)^{-1}} \leq \beta_k \frac{k + \lambda - g(k) - 8\sqrt{k \log(2dHk/\delta)}}{(k\lambda_h^+ + \lambda - g(k) - 8\sqrt{k \log(2dHk/\delta)})^{3/2}} = \widetilde{O}(k^{-1/2}),$$

*where $\lambda_h^+$ is the minimum nonzero eigenvalue of $\Lambda_h^\star$.*

*Proof.* We follow the proof scheme of Lemma 19 from [Papini et al., 2021]. Let $f(k) = g(k) + 8\sqrt{k \log(2dHk/\delta)} = \widetilde{O}(\sqrt{k})$. Notice that $f(k)$ is positive.

Fix $h$ and let $B_h^k = k\Lambda_h^\star + \lambda I - f(k)I$. First, notice that $B_h^k$ is an affine transformation of $\Lambda_h^\star$. As such, $B_h^k$ has the same orthonormal eigenvectors as $\Lambda_h^\star$, and we can define a mapping between the eigenvalues of the two matrices. Next, notice that $B_h^k$ is always invertible for sufficiently large $k$. Indeed, zero eigenvalues of $\Lambda_h^\star$ are mapped to negative eigenvalues of $B_h^k$ for sufficiently large $k$ — and since $f(k)$ is increasing and sublinear, positive eigenvalues of $\Lambda_h^\star$ are mapped to positive eigenvalues of $B_h^k$ for sufficiently large $k$. We call $\widetilde{\kappa}_h$ the smallest $k$ such as both conditions hold. For the rest of the proof assume $k \geq \kappa_h$. We have shown that $B_h^k$ is invertible and all and only the nonzero eigenvalues of $\Lambda_h^\star$ are mapped into positive eigenvalues of $B_h^k$, with the same orthonormal eigenvectors.

Now fix $(s, a)$ such that $\phi_h(s, a) \in \mathrm{span}\{\phi_h^\star(s) | \rho_h^\star(s) > 0\}$ and let $x = \phi_h(s,a)/\|\phi_h(s,a)\|$. From Lemma 30, with probability $1 - \delta$, $\Lambda_h^k \succeq B_h^k$. So:

$$x^\mathsf{T}(\Lambda_h^k)^{-1} x \leq x^\mathsf{T}(B_h^k)^{-1} x. \tag{101}$$

By hypothesis $x$ belongs to the column space $\mathrm{Im}(\Lambda_h^\star)$, so it belongs to the span of $\widetilde{d} \leq d$ orthonormal eigenvectors of $\Lambda_h^\star$. From the properties of $B_h^k$ stated above, $x$ belongs to the span of $\widetilde{d}$ orthonormal eigenvectors of $B_h^k$ corresponding to positive eigenvalues. The smallest such eigenvalue is:

$$k\lambda_h^+ + \lambda - f(k), \tag{102}$$

where $\lambda_h^+$ is the smallest nonzero eigenvalue of $M_h^\star$. Moreover, all the eigenvalues are upper bounded by:

$$k + \lambda - f(k). \tag{103}$$

From Proposition 31:

$$\|\phi_h(s,a)\|_{(\Lambda_h^k)^{-1}} \leq \sqrt{x^\mathsf{T}(\Lambda_h^k)^{-1} x} \tag{104}$$

$$\leq \sqrt{x^\mathsf{T}(B_h^k)^{-1} x} \tag{105}$$

$$\leq \frac{k + \lambda - f(k)}{k\lambda_h^+ + \lambda - f(k)} \frac{1}{\sqrt{x^\mathsf{T} B_h^k x}}. \tag{106}$$

Again from the properties of $B_h^k$, $x$ is orthogonal to all the orthonormal eigenvector of $B_h^k$ that correspond to zero eigenvalues of $\Lambda_h^\star$. Hence by Proposition 32:

$$x^\mathsf{T} B_h^k x = k x^\mathsf{T} \Lambda_h^\star x + \lambda - f(k) \tag{107}$$

$$\geq k \min_{y \in \mathrm{Im}(\Lambda_h^\star), \|y\|=1} y^\mathsf{T} \Lambda_h^\star y + \lambda - f(k) \tag{108}$$

$$= k\lambda_h^+ + \lambda - f(k). \tag{109}$$

Since $\beta_k = \widetilde{O}(1)$ and $f(k) = \widetilde{O}(\sqrt{k})$, from (106) and (109):

$$\beta_k \|\phi_h(s,a)\|_{(\Lambda_h^k)^{-1}} \leq \beta_k \frac{k + \lambda - f(k)}{(k\lambda_h^+ + \lambda - f(k))^{3/2}} = \widetilde{O}(k^{-1/2}). \tag{110}$$

$\square$

**Lemma 34.** *Let $\{\phi_j\}_{j\in[n]}$ be a set of $n$ vectors in $\mathbb{R}^d$ and $v \in \mathbb{R}^d$ be such that $v \notin \mathrm{span}\{\phi_j : j \in [n]\}$. Then, there exists a scalar $\epsilon > 0$ such that, for any $t \geq 0, \eta > 0$,*

$$\|v\|_{(t\sum_{j\in[n]} \phi_j\phi_j^T + \eta I)^{-1}} \geq \frac{\epsilon}{\sqrt{\eta}}.$$

*Proof.* Let $\{\lambda_i, u_i\}_{i\in[d]}$ denote the eigenvalues/eigenvectors of the matrix $\sum_{j\in[n]} \phi_j\phi_j^T$. Note that $\mathrm{span}\{u_i : i \in [d]\} = \mathrm{span}\{\phi_j : j \in [n]\} \subset \mathbb{R}^d$. Then, Lemma 28 of Papini et al. [2021] ensures that there exists a scalar $\epsilon > 0$ such that $|v^T u_i| \geq \epsilon$ for at least one eigenvector $u_i$ associated with a zero eigenvalue. Noting that the eigenvectors of $(t\sum_{j\in[n]} \phi_j\phi_j^T + \eta I)^{-1}$ are the same as the those of $\sum_{j\in[n]} \phi_j\phi_j^T$, we have that

$$\|v\|_{(t\sum_{j\in[n]} \phi_j\phi_j^T + \eta I)^{-1}}^2 = \sum_{j\in[d]} \frac{(v^T u_j)^2}{\eta + \lambda_j} \geq \frac{(v^T u_i)^2}{\eta} \geq \frac{\epsilon^2}{\eta},$$

which concludes the proof. $\square$

## G  Examples and Numerical Validations

Consider the following two-stage MDP ($H = 2$) with states $\mathcal{S} = \{s_1, s_2\}$ and actions $\mathcal{A} = \{a_1, a_2\}$:

$$r_1(s,a) = 1 \qquad \text{for all } s \in \mathcal{S} \text{ and } a \in \mathcal{A}, \tag{111}$$

$$p_1(s_1|s_1, a_1) = 1, \quad p_1(s_1|s_1, a_2) = \frac{1}{2}, \qquad p_1(s_1|s_2, a_1) = \frac{1}{2}, \quad p_1(s_1|s_2, a_2) = \frac{3}{4}, \tag{112}$$

$$r_2(s_1, a_1) = 1, \qquad r_2(s_1, a_2) = \frac{7}{8}, \qquad r_2(s_2, a_1) = \frac{1}{2}, \qquad r_2(s_2, a_2) = \frac{5}{8}, \tag{113}$$

$\mu(s_1) = \mu(s_2) = 1/2$, and of course $p(s_2|s,a) = 1 - p(s_1|s,a)$ for all $s \in \mathcal{S}$ and $a \in \mathcal{A}$. Backward induction shows that the (unique) optimal policy is:

$$\pi_1^\star(s_1) = a_1, \qquad \pi_1^\star(s_2) = a_2, \qquad \pi_2^\star(s_1) = a_1, \qquad \pi_2^\star(s_2) = a_2, \tag{114}$$

with the following values:

$$V_1^\star(s_1) = 2, \qquad V_1^\star(s_2) = \frac{61}{32}, \qquad V_2^\star(s_1) = 1, \qquad V_2^\star(s_2) = \frac{5}{8}. \tag{115}$$

Notice also that all states and actions are reachable, i.e. $\rho_h(s,a) > 0$ for all $s \in \mathcal{S}$, $a \in \mathcal{A}$, and $h \in [H]$.

**UNISOFT representation.** Consider the following 2-dimensional representation $\Phi^{(1)}$:

$$\underline{\phi_1^{(1)}(s_1, a_1)} = \begin{bmatrix} 1 \\ 0 \end{bmatrix} \quad \phi_1^{(1)}(s_1, a_2) = \begin{bmatrix} 1/2 \\ 1/2 \end{bmatrix} \quad \phi_1^{(1)}(s_2, a_1) = \begin{bmatrix} 1/2 \\ 1/2 \end{bmatrix} \quad \underline{\phi_1^{(1)}(s_2, a_2)} = \begin{bmatrix} 3/4 \\ 1/4 \end{bmatrix} \tag{116}$$

$$\underline{\phi_2^{(1)}(s_1, a_1)} = \begin{bmatrix} 0 \\ 1 \end{bmatrix} \quad \phi_2^{(1)}(s_1, a_2) = \begin{bmatrix} 1/4 \\ 3/4 \end{bmatrix} \quad \phi_2^{(1)}(s_2, a_1) = \begin{bmatrix} 1 \\ 0 \end{bmatrix} \quad \underline{\phi_2^{(1)}(s_2, a_2)} = \begin{bmatrix} 3/4 \\ 1/4 \end{bmatrix}. \tag{117}$$

It is easy to check that the MDP is low-rank (Asm 2) and $\Phi^{(1)}$ is a realizable representation with $\theta_1 = [1, 1]^T$, $\boldsymbol{\mu}_1(s_1) = [1, 0]^T$, $\boldsymbol{\mu}_1(s_2) = [0, 1]^T$, and $\theta_2 = [1/2, 1]^T$. This is an example of low-rank MDP with *simplex feature space* (see Example 2.2 in [Jin et al., 2020]). We have underlined optimal

features. It is easy to see that optimal features span $\mathbb{R}^2$ at both stages[9], so $\Phi^{(1)}$ is UNISOFT. The optimal covariance matrices are:

$$\Lambda_{1,\star}^{(1)} = \frac{1}{32}\begin{bmatrix} 25 & 3 \\ 3 & 1 \end{bmatrix}, \qquad\qquad \Lambda_{2,\star}^{(1)} = \frac{1}{128}\begin{bmatrix} 9 & 3 \\ 3 & 113 \end{bmatrix}. \qquad (118)$$

Both are full rank, and their minimum eigenvalues are:

$$\lambda_{1,+}^{(1)} = \frac{13 - 3\sqrt{17}}{32} \simeq 0.02, \qquad\qquad \lambda_{2,+}^{(1)} = \frac{61 - \sqrt{2713}}{128} \simeq 0.07. \qquad (119)$$

As shown in Theorems 8 and 9, both LSVI-UCB and ELEANOR will only suffer constant regret on this problem.

**Non-UNISOFT representation.** We apply the procedure described in the proof of Lemma 7 from [Papini et al., 2021] to the second stage of $\Phi^{(1)}$ to obtain an equivalent representation $\Phi^{(2)}$:

$$\phi_2^{(2)}(s_1, a_1) = \begin{bmatrix} 30/89 \\ 74/89 \end{bmatrix} \quad \phi_2^{(2)}(s_1, a_2) = \begin{bmatrix} 1/4 \\ 3/4 \end{bmatrix} \quad \phi_2^{(2)}(s_2, a_1) = \begin{bmatrix} 1 \\ 0 \end{bmatrix} \quad \phi_2^{(2)}(s_2, a_2) = \begin{bmatrix} 75/356 \\ 185/356 \end{bmatrix},$$
$$(120)$$

while the feature map for $h = 1$ is the same. It is easy to check that this is still a realizable representation for our MDP with the same parameters.[10] Although the UNISOFT property holds for $h = 1$, it no longer does for $h = 2$. Indeed, we have the following linear dependence between optimal features:

$$\phi_{2,\star}^{(2)}(s_2) = \frac{5}{8}\phi_{2,\star}^{(2)}(s_1), \qquad (121)$$

so optimal features only span $\mathbb{R}^1$. However, suboptimal features still span $\mathbb{R}^2$, e.g., by taking action $a_2$ in $s_1$ and $a_1$ in $s_2$ (recall that all state-action pairs are reachable). Due to Theorem 5, neither LSVI-UCB nor ELEANOR will achieve constant regret on this problem.

**Alternative Non-UNISOFT representation.** It is also easy to build a representation that is non-UNISOFT by changing the representation at the first stage. For example, let $h$ be any stage (e.g., $h = 1$ in our example) for which we want to transform a UNISOFT representation (in our case $\phi^{(1)}$) into a non-UNISOFT one. We can define a new representation $\phi^{(3)}$ as follows

$$s \in \mathcal{S}, \;\; \phi_h^{(3)}(s, a^\star) = \begin{bmatrix} 0_d \\ \phi_h^{(1)}(s, a_s^\star) \end{bmatrix} \qquad \forall a \neq a_s^\star, \;\; \phi_h^{(3)}(s, a) = \begin{bmatrix} \phi_h^{(1)}(s, a) \\ 0_d \end{bmatrix} \qquad (122)$$

$$s' \in \mathcal{S}, \;\; \mu_h^{(3)}(s') = \begin{bmatrix} \mu_h^{(1)}(s') \\ \mu_h^{(1)}(s') \end{bmatrix} \qquad\qquad\qquad \theta_h^{(3)} = \begin{bmatrix} \theta_h^{(1)} \\ \theta_h^{(1)} \end{bmatrix} \qquad (123)$$

Since all states are reachable, it is easy to verify that $\lambda_{\min}\left(\mathbb{E}_{s \sim \rho_h^\star}\left[\phi_h^{(3)}(s, a_s^\star)^\intercal \phi_h^{(3)}(s, a_s^\star)\right]\right) = 0$ and that

$$\mathrm{span}\Big\{\phi_h(s, a) \mid \forall(s, a), \; \exists \pi \in \Pi : \rho_h^\pi(s, a) > 0\Big\} \neq \mathrm{span}\Big\{\phi_h^\star(s) \mid \forall s, \; \rho_h^\star(s) > 0\Big\}.$$

Then, the representation is not UNISOFT at stage $h$.

### G.1 Numerical Validations

We provide a numerical validation of the behavior of the algorithms with and without a UNISOFT representation. We consider the following representations: $\phi^{(1)}, \phi^{(2)}, \phi^{(3)}$ which is obtained by applying the transformation in Eq. 122-123 to $\phi^{(2)}$ at stage $h = 1$, and $\phi^{(4)}$ which is obtained by

---

[9]It may appear counterintuitive that simplex features, which live on a one-dimensional manifold, can span $\mathbb{R}^2$. However, notice that the simplex is not a *linear* subspace of the Euclidean space (it does not include the origin). Indeed, we could describe the example MDP with less parameters, but we would loose the linear structure.

[10]However, notice that some of the new features do not belong to the simplex.

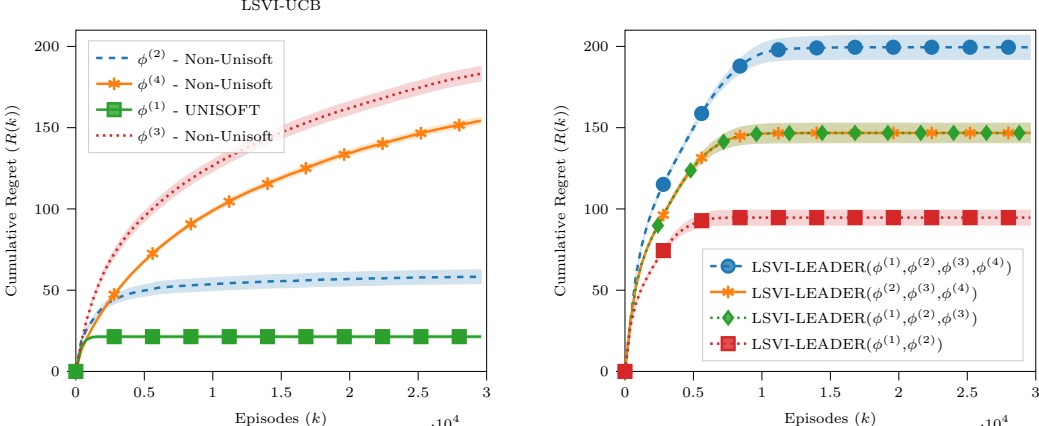

Figure 1: Cumulative regret of LSVI-UCB and LSVI-LEADER with different representations. The performance of LSVI-LEADER with $\{\phi^{(1)}, \phi^{(2)}, \phi^{(4)}\}$ is the same of the one with $\{\phi^{(1)}, \phi^{(2)}, \phi^{(3)}\}$ and $\{\phi^{(2)}, \phi^{(3)}, \phi^{(4)}\}$.

applying the transformation in Eq. 122-123 to $\phi^{(1)}$ at stage $h = 1$. Note that we have $d_1 = 4$ and $d_2 = 2$ for $\phi^{(3)}$ and $\phi^{(4)}$. Furthermore, $\lambda_{1,1}^{(3)} = \lambda_{1,1}^{(4)} = 0$, while $\lambda_{2,1}^{(3)} = 0$ and $\lambda_{2,1}^{(4)} > 0$, which means that $\phi^{(4)}$ is "locally" UNISOFT at stage $h = 2$. The reward is stochastic and drawn from a Bernoulli distribution: $r_{h,t} \sim \text{Ber}(r_h(s_t, a_t))$. We tested both LSVI-UCB on each individual representation and LSVI-LEADER with different combinations of the representations. We consider $\beta_{h,k} = c_\beta d_h H \sqrt{\log(d_h K)}$ and $\beta_{h,k} = c_\beta d_h H \sqrt{N \log(N d_h K)}$ for LSVI-UCB and LSVI-LEADER, respectively. We set $c_\beta = 0.2$ and $K = 30000$. The regret is shown in Fig. 1, averaged over the same 100 seeds.

As expected from the theoretical analysis, LSVI-UCB with UNISOFT representation suffers constant regret since, after the initial exploration phase, it only selects optimal actions. On the other hand, when the representation is Non-UNISOFT, LSVI-UCB suffers a non-constant regret that grows over episodes. LSVI-LEADER is able to exploit the structure of the UNISOFT representation and it achieves constant regret as well in all the configurations containing a UNISOFT representation. The higher regret is due to a longer exploration phase that is a consequence of the enlarged confidence intervals; this is also in line with the theoretical analysis. It is interesting to notice that LSVI-LEADER performs equally good with all the combinations of representations of dimension three (i.e., $\{\phi^{(1)}, \phi^{(2)}, \phi^{(4)}\}$, $\{\phi^{(1)}, \phi^{(2)}, \phi^{(3)}\}$ and $\{\phi^{(2)}, \phi^{(3)}, \phi^{(4)}\}$). LSVI-LEADER is indeed able to mix representations and achieve constant regret even when none of the individual representation would. In the case of $\{\phi^{(2)}, \phi^{(3)}, \phi^{(4)}\}$, LSVI-LEADER is able to mix $\phi^{(2)}$ and $\phi^{(4)}$, that are UNISOFT in stage $h = 1$ and $h = 2$, respectively.

**UNISOFT in DeepRL.** We wanted also to verify the existence of UNISOFT representations in DeepRL. We trained A2C [Mnih et al., 2016] on different domains and evaluated whether the recovered representation (i.e., last layer of the neural network used to approximate $V^\star$) satisfies the UNISOFT assumptions. Standard benchmark problems are not finite-horizon, we thus considered the following "strong" UNISOFT condition $\lambda_{\min}\left(\mathbb{E}_{s \sim \rho^\star}[\phi^\star(s)\phi^\star(s)^\mathsf{T}]\right) > 0$, which was evaluated by simulating multiple trajectories:

$$\Lambda_m^\pi = \frac{1}{m} \sum_{i=1}^{m} \sum_{t=1}^{T_i} \phi(s_t, a_t)\phi(s_t, a_t)^\mathsf{T} \tag{124}$$

where $a_t = \pi(s_t)$. We use a deterministic version of the policy recovered by A2C for evaluation. We trained A2C using the implementation provided by stable-baselines3 [Raffin et al., 2019]. We use the default parameters (provided by rl-baselines3-zoo [Raffin, 2020]) and tested different network architectures. Since we did not optimize the parameter, we reported only the domains where we obtained good results with at least one network architecture (highlighted in the table). Since A2C

| Domain | mean reward | std reward | eval timesteps | eval episodes ($m$) | rank($\Lambda_m^\pi$) | $\lambda_{\min}(\Lambda_m^\pi)$ | UNISOFT |
|---|---|---|---|---|---|---|---|
| Acrobot-v1 | -84.5 | 20.7 | 149923 | 1753 | 16 | 0.02 | ✔ |
| AntBulletEnv-v0 | 2303.9 | 68.3 | 150000 | 150 | 15 | 0 | |
| BipedalWalker-v3 | 2.2 | 1.6 | 148800 | 93 | 10 | 0 | |
| CartPole-v1 | 500.0 | 0.0 | 150000 | 300 | 1 | 0 | |
| HopperBulletEnv-v0 | 836.3 | 536.2 | 149982 | 372 | 16 | 0 | |
| MountainCar-v0 | -124.9 | 31.4 | 149979 | 1201 | 16 | 0.01 | ✔ |
| MountainCarContinuous-v0 | 91.6 | 0.2 | 149966 | 1736 | 5 | 0 | |
| Pendulum-v0 | -173.5 | 107.0 | 150000 | 750 | 16 | 0 | |

Table 3: Results for A2C policy network of dimension $[64, 16]$ and value network of dimension $[64, 16]$. These dimensions represent the size of the hidden layers (with tanh activation function). We highlighted the environments where A2C achieved good performance.

| Domain | mean reward | std reward | eval timesteps | eval episodes ($m$) | rank($\Lambda_m^\pi$) | $\lambda_{\min}(\Lambda_m^\pi)$ | UNISOFT |
|---|---|---|---|---|---|---|---|
| Acrobot-v1 | -84.9 | 29.4 | 149987 | 1747 | 32 | 0.0018 | ✔ |
| AntBulletEnv-v0 | 2109.9 | 46.1 | 150000 | 150 | 32 | 0.0010 | ✔ |
| BipedalWalker-v3 | 267.3 | 53.3 | 149278 | 201 | 24 | 0 | |
| CartPole-v1 | 500.0 | 0.0 | 150000 | 300 | 1 | 0 | |
| HopperBulletEnv-v0 | 1461.6 | 707.1 | 149123 | 205 | 32 | 0.0001 | ✔ |
| MountainCar-v0 | -116.5 | 28.0 | 149999 | 1288 | 32 | 0.0001 | ✔ |
| MountainCarContinuous-v0 | 91.5 | 0.2 | 149975 | 1742 | 10 | 0 | |
| Pendulum-v0 | -236.5 | 187.7 | 150000 | 750 | 26 | 0 | |

Table 4: Results for A2C policy network of dimension $[64, 32]$ and value network of dimension $[64, 32]$. These dimensions represent the size of the hidden layers (with tanh activation function). We highlighted the environments where A2C achieved good performance.

estimates directly $V^\star$, we used the features of the last layer as features of the optimal policy (i.e., $\phi^\star(s)$) to test for the "strong" UNISOFT condition.

Tables 3–5 show that in several domains the learnt representation is UNISOFT, although the minimum eigenvalue is small. As expected, the number of "strong" UNISOFT representations decreases as the size of the last layer increases. This initial experiment shows that UNISOFT representations are not uncommon in practice but also leave open the possibility of designing algorithms that explicitly try to force the UNISOFT while learning. We believe this is an interesting direction for future work.

| Domain | mean reward | std reward | eval timesteps | eval episodes ($m$) | rank($\Lambda_m^\pi$) | $\lambda_{\min}(\Lambda_m^\pi)$ | UNISOFT |
|---|---|---|---|---|---|---|---|
| Acrobot-v1 | -83.3 | 17.1 | 149970 | 1778 | 64 | 0.0003 | ✔ |
| AntBulletEnv-v0 | 1912.7 | 106.0 | 150000 | 150 | 64 | 0.0008 | ✔ |
| BipedalWalker-v3 | 276.1 | 25.8 | 149707 | 198 | 28 | 0 | |
| CartPole-v1 | 500.0 | 0.0 | 150000 | 300 | 2 | 0 | |
| HopperBulletEnv-v0 | 14.0 | 0.8 | 149997 | 26620 | 59 | 0 | |
| MountainCar-v0 | -107.3 | 20.1 | 149944 | 1397 | 44 | 0 | |
| MountainCarContinuous-v0 | 92.4 | 0.1 | 149984 | 1948 | 10 | 0 | |
| Pendulum-v0 | -153.3 | 92.9 | 150000 | 750 | 32 | 0 | |

Table 5: Results for A2C policy network of dimension $[64, 64]$ and value network of dimension $[64, 64]$. These dimensions represent the size of the hidden layers (with tanh activation function). We highlighted the environments where A2C achieved good performance.