# OpenReview forum: "Reinforcement Learning in Linear MDPs: Constant Regret and Representation Selection"
_NeurIPS.cc/2021/Conference — NeurIPS 2021 Poster_

### Official Review · Reviewer_rJnu · 2021-07-16

**Rating:** 5
**Confidence:** 4

**Summary:**

This paper studies representation learning in reinforcement learning. The authors prove that the proposed UNI-SOFT assumption is sufficient and necessary for achieving the constant regret bound for a single representation. The authors also propose an algorithm to select the representation to achieve the constant regret bound when any single representation cannot achieve this.

**Limitations And Societal Impact:**

No negative societal impact of this work found. Here's my suggestions

It is still not clear to me how to verify the UNI-SOFT condition. It would help readers understand the performance of the representation learning algorithm by adding some experiments.

**Main Review:**

The paper is overall well organized and easy to follow. Despite the aforementioned contributions, there're still several drawbacks in this paper.

First, given the previous logarithmic regret bound given by [He et. al, 2020] and the constant regret bound for linear contextual bandit given by [Papini et. al, 2021], the constant regret bound for linear MDP is somewhat easy to obtain. In detail, as the authors show in line 153, the UNI-SOFT assumption is similar to the HLS condition in [Papini et. al, 2021]. They both show that the representation from the optimal action can provide rich information on the whole space of representation. Given this similarity, the proof of the constant regret bound is quite similar with [Papini et. al, 2021]

Second, regarding the representation selection part, the authors acknowledge in line 294 that there's a $\sqrt{N}$ inflation in the confidence radius $\beta$. However, this additional $\sqrt{N}$ is crucial for representation learning. According to [He et. al, 2020], this additional $\sqrt{N}$ factor will lead to an additional $N$ dependency on the final regret bound. This additional $N$ dependency will make the result infeasible when the representation candidate set is large.

Therefore, due to the drawbacks mentioned above, especially the crucial additional $N$ factor in regret bound for representation selection, I would like to suggest marginally reject this paper.

[1] Papini, Matteo, et al. "Leveraging good representations in linear contextual bandits." International Conference on Machine Learning. PMLR, 2021.

[2] He, Jiafan, Dongruo Zhou, and Quanquan Gu. "Logarithmic regret for reinforcement learning with linear function approximation." International Conference on Machine Learning. PMLR, 2021.

**Time Spent Reviewing:**

6

---

> ### Author Response · Authors · 2021-08-10
> **Answer to reviewer rJnu**
>
> We thank the reviewer for the feedback and for the time spent reviewing our paper.
>
> > the constant regret bound for linear MDP is somewhat easy to obtain.
>
> The multi-stage nature of RL poses some non-trivial, albeit subtle, challenges. For instance, it is not obvious a priori why the UNISOFT assumption should have that form (e.g. why should the condition hold separately for each stage $h$?). It is only through our proofs of necessity and sufficiency that we show that UNISOFT is indeed the correct generalization. Indeed, the proofs carry some surprises: for instance, the form of optimism that is required to exploit UNISOFT (Asm. 16) is quite peculiar and we could have not guessed it in advance. See also the bullet points in the answer to Reviewer Nzqf.
>
> > According to [He et. al, 2020], this additional $\sqrt{N}$ factor will lead to an additional $N$ dependency on the final regret bound.
>
> Indeed, if the gap-dependent regret by He et al. is used in Theorem 10, it leads to a linear dependence on $N$. We thank the reviewer for pointing this out. However, the linear dependence can be avoided by using the worst-case bound by Jin et al, that leads to a $\sqrt{N}$ dependence. We will make this clear in the final version.
>
> > It would help readers understand the performance of the representation learning algorithm by adding some experiments.
>
> See answer to Nzqf above, and the example from Appendix F.

---

> > ### Author Response · Authors · 2021-08-26
> > **Has our response addressed your concerns?**
> >
> > Hello reviewer rJnu, we would be grateful if you can confirm whether our response has addressed your concerns, and let us know if any issues remain.
> > In addition to the original answer:
> > - We provided a detailed description of the technical challenges and novelty.
> > - We provided a numerical simulation validating the theoretical findings. In particular, the experiment shows: i) the importance of the UNISOFT assumption, ii) our algorithms can leverage the UNISOFT structure and achieve constant regret.
> >
> > Thank you in advance.

---

> > > ### Comment · Reviewer_rJnu · 2021-08-26
> > > **Experiment setting questions**
> > >
> > > Thank you for your response, I still have some questions regarding the experiment. In Appendix F, it seems that $H = 2$. In that case, the representation $\phi_2$ is only used when calculating the reward function instead of the transition kernel. Thus, it seems that the environment setting is more like a bandit setting. So, I'm wondering could the UNISOFT condition be applied to the representation which affects the transition kernel $\phi_1$ here. Is it possible to provide examples on how to get an UNISOFT $\phi_1^{(1)}$ and a non-UNISOFT $\phi_1^{(2)}$ in your case?

---

> > > > ### Author Response · Authors · 2021-09-02
> > > > **Answer**
> > > >
> > > > The reviewer is right, in the example in appendix F we get a non-unisoft representation $\phi^{(2)}$ by changing the last stage that has no associated transitions. It is easy to build a representation that is non-UNISOFT by changing the representation at the first stage.
> > > >
> > > > For example, let $h$ be any stage (e.g., $h=1$ in our example) for which we want to transform a UNISOFT representation (in our case $\phi^{(1)}$) into a non-UNISOFT one. We can define a new representation $\phi^{(3)}$ as follows
> > > >
> > > > - $\phi^{(3)}_h(s,a^\star_s) = [0_d,\phi_h^{(1)}(s,a^\star_s)]$ for any $s$
> > > > - $\phi^{(3)}_h(s,a) = [\phi_h^{(1)}(s,a),0_d]$ for any $s$, $a \neq a^\star_s$
> > > > - $\mu^{(3)}_h(s’) = [\mu_h^{(1)}(s’),\mu_h^{(1)}(s’)]$
> > > > - $\theta^{(3)}_h = [\theta_h^{(1)},\theta_h^{(1)}]$
> > > >
> > > > It is easy to verify that $\lambda_{\min}(\mathbb{E}_{s \sim \rho^\star_h}[\phi^{(3)}_h(s,a^\star_s))^\intercal \phi^{(3)}_h(s,a^\star_s)]) = 0$.

---

> > > > > ### Author Response · Authors · 2021-09-02
> > > > > **Has our response addressed your concerns?**
> > > > >
> > > > > Hello again reviewer rJnu, we would be grateful if you can confirm whether our responses has addressed your concerns, and let us know if any issues remain.
> > > > >
> > > > > Thank you in advance.

---

### Official Review · Reviewer_nyW6 · 2021-07-16

**Rating:** 7
**Confidence:** 3

**Summary:**

This paper introduces the “UNISOFT” assumption which roughly states that the features for the state-action pairs visited by the optimal policy must span the entire feature space. A lower bound is given which shows that the assumption is necessary to achieve constant regret, and upper bounds are given which establish that the LSVI-UCB and ELEANOR algorithms achieve constant regret when UNISOFT holds (but also with a gap condition as well as their respective linear MDP and low IBE assumptions). The authors also analyze the representation selection problem (where a candidate set of features is given) and propose the LSVI-LEADER algorithm which has regret as small as that of the best feature, and in particular constant regret if any of those features is UNISOFT.

**Limitations And Societal Impact:**

As mentioned above: I think certain technical points should be clarified and/or corrected and the writing should be improved in several places.

**Main Review:**

From a technical standpoint this seems like a strong paper with interesting technical contributions towards the problem of low-regret algorithms for linear function approximation (although under strong assumptions). For this reason I am recommending acceptance (albeit a weak one for now for reasons that I will be discussing shortly).

The proofs are neat and mostly seem to check out, although I did not check them in great detail. I particularly enjoyed the lower bound (Theorem 5 and Lemma 7). I think the upper bound results would also be strengthened by emphasizing that, with UNISOFT, constant regret is achieved for any algorithm that falls under the very standard form of “optimism + elliptical confidence bounds”.

There are a number of confusing or potentially concerning points, however. The main claim is that UNISOFT gives constant regret, but it is very unclear from the stated bounds (Theorems 8 and 9) how the regret is independent of K. In particular the bounds scale with \bar{\kappa} (defined in the text as “the last episode where ELEANOR suffers non-zero regret”), which I would be very surprised to learn is independent of K. In particular If ELEANOR suffered non-zero regret for a number of episodes independent of K then surely ELEANOR would also have constant regret? (e.g.  an easy bound would be Regret <= \bar{\kappa} * H). The text states “As expected, \bar{\kappa} is independent of the number of episodes K” as if it were evident, so it is possible that I am missing something obvious here. Either way, I think some clarification on the K-independence of this bound is in order. Examining the proof for this is not illuminating either — the K-independence comes from Eqn (3) of the sketch (or Eqn (28) of the full proof), and it is not clear how the authors arrive at these crucial inequalities (the <= \Delta_min bound). Can the authors elaborate on this?

Another technical concern is about the lower bound. The text after Theorem 5 states that the class of MDPs given in the Theorem statement (i.e. with linear rewards and arbitrary dynamics) subsumes low-rank MDPs, linear-mixture MDPs, and MDPs with Bellman closure. The last of these claims is untrue — linear rewards alone do not guarantee Bellman closure as far as I can tell (due to the max over actions being inside the expectation, the update T(q) can not be written as a single linear function). If this is indeed not true then the lower bound does not actually show that UNISOFT is necessary for constant regret in Bellman-closed MDP-feature-map pairs, so the results are not as strong as they claim. On a related note: the Bellman optimality equation which is given in the Preliminaries section is incorrectly stated (!)

There are some writing concerns which I also think should be addressed. The intro sets up an unclear dichotomy between “good” representations and “learnable” representations. These terms aren’t defined very well (except circularly), and the rest of the paper uses this language in a technical sense (cf. Assumption 4). If we take the meaning expressed in Assumption 4 as the technical definition then it seems that learnable features are defined exactly as the features of Low-Rank MDPs or features with low Bellman Error, which seems like a restrictive definition for such a general idea. Furthermore, if a “good” representation is one that is realizable and a “learnable” definition is as above then one of these definitions is a strict subset of the other one. In summary I do not think this dichotomy adds anything and just confuses the narrative, and I would advise revising this section of the paper. On a more minor note, there is an inconsistency in the text about whether the term “linear MDP” is used (as in the title) or “low-rank MDP” (as in Assumption 2). I think the UNISOFT condition should also be loosely introduced in the setup of the paper, rather than saying “good” in the abstract.

Lastly, a question for the authors: why does unisoft need to assume a unique optimal action? Does this come into the proof somewhere?


**Time Spent Reviewing:**

6

---

> ### Author Response · Authors · 2021-08-10
> **Answer to reviewer nyW6**
>
> We thank the reviewer for the positive feedback and for the time spent reviewing our paper.
>
> > I think the upper bound results would also be strengthened by emphasizing that, with UNISOFT, constant regret is achieved for any algorithm that falls under the very standard form of “optimism + elliptical confidence bounds”.
>
> Thanks for the suggestion, indeed this is expressed by Theorem 19 in the appendix. We will highlight this in the main paper as well.
>
> > it is very unclear from the stated bounds (Theorems 8 and 9) how the regret is independent of K.
>
> The confusion arises from the fact that we do not have a compact expression for the critical time $\kappa$, which is really a constant independent of the horizon K. However, by analyzing eq. 27 in the proof of Thm. 19, we can show that $\overline{\kappa} = \tilde{O}(d^2H^4\Delta^{-2}\lambda_+^{-3})$, where $\lambda_+=\min_{h}\lambda_h^+$ and $\tilde{O}$ in this case hides logarithmic terms in $d$, $H$, $\Delta$ and $\lambda_+$ (but not in $K$). So $\overline{\kappa}$ is really independent of $K$ and has polynomial dependence on other problem-dependent constants. We shall report this characterization of $\overline{\kappa}$ in the final version.
>
> The proof sketch of Sec. 3.2 should clarify why this is possible. In short, we prove that under UNISOFT these algorithms only play optimal actions after $\overline{\kappa}$. So, as the reviewer correctly observed, the regret is bounded by $\overline{\kappa} H$. Since $\kappa$ still depends on $d$, $H$, $\Delta$ and $\lambda_+$, it is better to plug $\overline{\kappa}$ into existing logarithmic upper bounds to obtain a more favorable dependence w.r.t. these constants.
>
> As for the proof, in (28) we just solve the inequality for k, obtaining $\kappa$. Here it is enough to observe that a finite $\kappa$ must exist since the $(\lambda^+ k)^{3/2}$ terms in the denominator of (27) dominate, making the instantaneous regret shrink and eventually fall under $\Delta_{\min}$. Note that $K$ does not appear anywhere in this inequality. Finally, the gap assumption implies that the instantaneous regret is zero after $\kappa$. We will make this more clear in the proof sketch too.
>
> > linear rewards alone do not guarantee Bellman closure
>
> Indeed, it is the other way around. But Thm 5 is a negative result: whenever there are linear rewards, UNISOFT is necessary for constant regret. Since Bellman closure implies linear rewards (Prop. 2 by Zanette et al., 2020), the theorem applies to Bellman closure as well.
>
> > the Bellman optimality equation which is given in the Preliminaries section is incorrectly stated (!)
>
> Thanks for pointing out that typo. It has already been fixed.
>
> > There are some writing concerns which I also think should be addressed.
>
> Thanks for the suggestions: we will just use UNISOFT and low-rank MDPs in technical parts.
>
> > why does unisoft need to assume a unique optimal action? Does this come into the proof somewhere?
>
> The assumption is used in the very definition of UNISOFT. The case of multiple optimal actions is significantly more difficult to analyze because the set of “optimal features” depends on the particular optimal policy the algorithm converges to. In summary, to get constant regret with UNISOFT we first need to show that the algorithm converges to playing certain optimal actions, and then use the fact that the features of such actions span the whole space. Unfortunately, it is an open question to characterize what policy existing optimistic algorithms converge to (and whether they converge in the first place) in the presence of multiple optimal actions.

---

> > ### Author Response · Authors · 2021-08-26
> > **Has our response addressed your concerns?**
> >
> > Hello reviewer nyW6, we would be grateful if you can confirm whether our response has addressed your concerns, and let us know if any issues remain.
> >
> > Thank you in advance.

---

> > > ### Comment · Reviewer_nyW6 · 2021-08-31
> > > **Thanks for the response**
> > >
> > > Hi there, apologies for the late reply on my behalf.
> > >
> > > Thanks for addressing my concerns — the K-independence of the regret bound is clearer to me now. One follow up question that I have is whether the characterization \bar{\kappa} = O(d^2H^4\Delta^{-2}\lambda^{-3}) always holds or if this depends on the UNISOFT condition?
> > >
> > > In the meantime, since you are planning to clarify the text a bit more, I am happy to increase my score to a full accept.

---

> > > > ### Author Response · Authors · 2021-09-02
> > > > **Answer**
> > > >
> > > > This bound holds for a UNISOFT representation (Assumption 4). However, note that we can always take the minimum between $\kappa$ and $K$, thus recovering the standard regret bound for non-UNISOFT representations.

---

### Official Review · Reviewer_Nzqf · 2021-07-17

**Rating:** 5
**Confidence:** 3

**Summary:**

The paper considers problem dependent bounds for finite horizon MDP. The problem dependent regret bound do not have a logarithmic term of time, which is the key innovation as compared to the prior works.

**Limitations And Societal Impact:**

yes

**Main Review:**

The paper gives the first results where log(T) factor could be removed in problem dependent setup of MDPs. Even though the result is new, the usefulness of the result is unclear. Since there is already a \sqrt{T} term for the regret, saving O(\log T) has a negligible performance improvement in realistic setup. The paper also lacks any numerical result to explain the use of this result in practice. The key novelty in the proof also needs better explanation.



**Time Spent Reviewing:**

6

---

> ### Author Response · Authors · 2021-08-10
> **Answer to reviewer Nzqf**
>
> We thank the reviewer for the feedback and for the time spent reviewing our paper.  There seems to be a significant misunderstanding about our results that we now seek to clarify.
>
> > the usefulness of the result is unclear. Since there is already a \sqrt{T} term for the regret, saving O(\log T) has a negligible performance improvement in realistic setup.
>
> There is no $\sqrt{T}$ term in the regret bound. As stated in Theorem 8 and 9, the regret is really constant under the UNISOFT assumption. We are not just removing the logarithmic terms from the $\sqrt{T}$ bound by Jin et al, we are replacing ALL terms that depend on T with the critical time $\tau$, that is finite and independent of the horizon T under the UNISOFT assumption.
>
> > The paper also lacks any numerical result to explain the use of this result in practice.
>
> The focus of the paper is on providing a full characterization of “good” representation for linear representations in MDPs. In particular, we show an intrinsic separation between the regret bound we can achieve with UNISOFT or not (UNISOFT is necessary for obtaining constant regret). While this already shows the importance of this condition, we will try to validate the findings through numerical simulations.
>
> > The key novelty in the proof also needs better explanation.
>
> The proof strategy to obtain constant regret is inspired by Papini et al. 2021, a contextual bandit paper. The novelty is in handling the multi-stage nature of the RL problem. In particular,
> * the proof that UNISOFT is necessary for constant regret builds on top of a novel lower bound for regret minimization in linear MDPs that we derive in Lemma 7 and which could be of independent interest (while the proof of Papini et al. 2021 builds on top of existing results from the linear bandit literature);
> * the proof of the constant regret upper bounds for ELEANOR and LSVI-UCB involves some technical results which are not common in the linear MDP literature (and which are quite challenging to extend from the bandit setting), including characterizing the growth in the eigenvalues of the design matrices (Lemma 34);
> * both the design and analysis of LSVI-LEADER involve handling the combination of multiple representations in a non-trivial way in order to show strong mixing results;
>
> We will add some comments about this in the main paper.
>
> > In addition, recently (Jin et al., 2021) proposed a new class of Bellman Eluder Dimension, and it would be interesting to comment if the results would hold for this class too?
>
> From a preliminary investigation, it seems that Bellman Eluder Dimension reduces to Bellman closure in the case of linear parameterizations. If so, our results will apply also to such a scenario. We don’t know at this time if the results would hold for non-linear representations. It is definitely an interesting question and we will look into it more closely after the rebuttal.

---

> > ### Comment · Reviewer_Nzqf · 2021-08-15
> > **Response**
> >
> > Thanks for providing the response.
> >
> > The authors responded to most of the misunderstanding of the comments. Some numerical validation into the assumptions satisfaction, and its impact would be helpful. Also, some numerical examples to demonstrate this regret will be nice. Would be good to see how the authors will address the novelty in the revised version, the written statement in rebuttal is concise.
> >
> > Based on the response, I am increasing the score.

---

> > > ### Author Response · Authors · 2021-08-26
> > > **Has our response addressed your concerns?**
> > >
> > > Hello reviewer Nzqf, we would be grateful if you can confirm whether our new responses have addressed your concerns, and let us know if any issues remain.
> > > To recap our response:
> > > - We provided a numerical simulation validating the theoretical findings. In particular, the experiment shows that our algorithms can leverage the UNISOFT structure and achieve constant regret.
> > > - We provided a detailed description of the technical challenges and novelty. The work in this paper is not simply a combination of results in the literature, but required original solutions that, on several occasions, pushed the theoretical understanding of the linear MDP framework.
> > >
> > > Thank you in advance.

---

### Official Review · Reviewer_UvSm · 2021-07-19

**Rating:** 7
**Confidence:** 4

**Summary:**

In this paper, the authors study conditions for achieving constant regret in RL with linear function approximation. They show that under a condition called UNISOFT, which says that features of state-actions pairs that can be reached by the optimal policy has the same span as features of state-action pairs that can be reached by any policy, constant regret is achievable if additional representational condition (completeness or linear MDP) holds. The authors also show that LSVI-LEADER (a modified version of LSVI-UCB), when given a set of N different representations, can adapt to the best representation automatically and achieve a regret bound that is upper bound by that of the best representation. Therefore, LSVI-LEADER achieves constant regret if any of the given representation satisfies UNISOFT.

**Limitations And Societal Impact:**

see main review.

**Main Review:**

This paper studies conditions that permit constant regret in RL with linear function approximation. The UNISOFT condition is intuitive, and the authors prove that such a condition is necessary for achieving constant regret bound. Moreover, if additional representation conditions that permit sample-efficient RL with linear function approximation hold, UNISOFT is also sufficient for achieving constant regret. Therefore, the authors give an almost complete characterization of conditions that permit constant regret in the linear function approximation setting.

The authors then discuss how to select a representation that satisfies UNISOFT in linear MDPs. In particular, they propose an algorithm that receives M different representations as input and show that the regret of the algorithm is upper bounded by that of the best representation. Therefore, if any of the given M representation satisfies UNISOFT, the algorithm achieves constant regret. Algorithmically, the idea is to solve M different linear regression problems that share the same set of targets, and set the estimated Q value to be the smallest one among all the M predictions. The algorithm itself is a generalization of the one in Papini et al. which works in the bandit setting. However, it should be noted that the representation condition used here is rather strong: it requires the transition probabilities to be linear with respect to all the given M representations, which could be restrictive.

In summer, this paper studies an interesting question. The authors give an almost complete characterization of conditions that permit constant regret in the linear function approximation setting. The authors also discuss how to automatically utilize a set of representation which potentially contains one representation that satisfies UNISOFT. This paper is also well written. Overall, this is a good paper and therefore I recommend acceptance.

**Time Spent Reviewing:**

10

---

> ### Author Response · Authors · 2021-08-10
> **Answer to reviewer UvSm**
>
> We would like to thank the reviewer for the positive feedback and for the nice words about the interest of our contributions.
>
> > it should be noted that the representation condition used here is rather strong: it requires the transition probabilities to be linear with respect to all the given M representations, which could be restrictive.
>
> The condition is indeed strong and we plan to look into multiple misspecified representations in future work. However, the current setting can be seen as a theoretical proxy of the realistic setting where different representations are learned from historical data. For example, if we train large (over-parameterized) neural networks, it is reasonable to expect to recover realizable representations.

---

### Author Response · Authors · 2021-08-23
**Technical challenges and novelty**

As kindly requested by reviewers, we provide more details on the technical challenges we had to face in solving the problems studied in the paper. We hope this will help the committee appreciate the elements of novelty in our work, in particular w.r.t. previous works on contextual bandits (Papini et al. 2021) and low-rank MDPs (Jin et al. 2020, Zanette et al. 2020, He et al. 2020) that provided inspiration and fundamental building blocks for our theoretical analysis.

- Unisoft is necessary (Section 3.1/Appendix B): our proof of Theorem 5 is based on a lower bound for regret minimization in linear MDPs (Lemma 7). This approach is inspired by the proof of Proposition 2 by Papini et al. (2021) for the case of contextual bandits. However, the latter was based on an existing regret lower bound for contextual bandits by Hao et al. (2020), which does not admit an easy generalization to MDPs. Instead, our Lemma 7 is a novel lower bound for linear MDPs that was technically challenging to obtain and may also be of independent interest.

- Unisoft is sufficient (Section 3.2/Appendix C): our proof of Theorem 19 (of which Thm 8 and 9 are special cases) follows the logic of the proof of Lemma 2 by Papini et al. (2021) on contextual bandits. The latter consists of two key steps: (1) bounding the instantaneous regret with Mahalanobis feature norms and (2) showing that these norms are decreasing under the UNISOFT assumption. Both steps raised additional technical challenges once applied to linear MDPs.
  1. Compared to contextual bandits, in linear MDPs the instantaneous regret has a more complex (recursive) relationship with feature norms, as first observed by Jin et al. (2020, Lemma B.6). We could only handle this additional challenge by deriving a novel upper bound on the gaps, that is our Lemma 18. The novelty of this result is apparent from the most general condition under which it holds (Assumption 16). “Almost local optimism” is a new algorithmic concept that is trivially satisfied by LSVI-UCB, but also less obviously by ELEANOR, for which Zanette et al. (2020) only established the weaker property of “global optimism”.
  2. Identifying the correct form of the UNISOFT assumption (i.e., the correct generalization of the HLS property by Papini et al., 2021) was crucial for establishing this result, and not a trivial task. The challenge was in correctly defining “optimal features” for the multi-step case, which turned out to be the ones observed in trajectories generated by the optimal policy. To see why leveraging the contribution of these features is technically more challenging than in the bandit case, please refer to the proof of Lemma 28, where a careful study of trajectory distributions was necessary.

- Representation selection (Section 4/Appendix D): LSVI-LEADER is conceptually similar to the LEADER algorithm by Papini et al. (2021), however its analysis comes with some additional technical challenges. As first observed by Jin et al. (2020, Lemma D.6), studying multi-step error propagation requires a precise characterization of the functional class of optimistic value functions, in terms of its covering number. The main challenge here was in handling the complex value function used by LSVI-LEADER, which is built by combining N representations (Lemma 20). This is the technical challenge that ultimately generated the polynomial dependence on N in Thm 10. We are looking into improving such dependence, but it remains an open question at the moment.

To sum up, applying ideas from contextual bandits to linear MDPs was not just a matter of combining results from the two literatures, but required original solutions that, on several occasions, pushed the theoretical understanding of the linear MDP framework.

---

### Author Response · Authors · 2021-08-24
**Numerical validation**

As requested by Reviewer Nzqf and Reviewer rJnu, we have implemented the approach and provided a first numerical validation on the behavior of LSVI-UCB with a Unisoft representation. We have considered the example reported in Appendix F, where representation $\phi^{(1)}$ is UNISOFT while $\phi^{(2)}$ is Non-UNISOFT. We tested both LSVI-UCB on each individual representation	 and LSVI-LEADER with both representations. The regret is shown in the figure [[link]](https://1drv.ms/u/s!AqqW3UTO_gd-bp03vsw8PSMkiL4), averaged over 15 runs.

As expected from the theoretical analysis, LSVI-UCB with UNISOFT representation $\phi^{(1)}$ suffers constant regret since, after the initial exploration phase, it only selects optimal actions. On the other hand, when the representation is Non-UNISOFT, LSVI-UCB suffers a non-constant regret that grows over episodes.
LSVI-LEADER is able to exploit the structure of the UNISOFT representation and it achieves constant regret as well. The higher regret is due to a longer exploration phase that is a consequence of the enlarged confidence intervals; this is also in line with the theoretical analysis.

We believe to have addressed your request by providing a preliminary empirical investigation of the impact of UNISOFT representations on the empirical performance. Our results provide empirical support to the theoretical findings.

---

### Decision · Program_Chairs · 2021-09-28

**Decision:**

Accept (Poster)

**Comment:**

The paper studies "fine-grained" properties of linear representations in reinforcement learning, in particular when such a representation can admit constant regret guarantees, going beyond the worst case bounds of \sqrt{T}-type. The paper also consider a representation selection setting, where one representation among a class admits a constant-regret bound.

The paper is a natural generalization of recent work on fine-grained representation properties in the linear bandits literature, but we believe it is a noteworthy contribution, so we are recommending that the paper is accepted to the conference.

A couple of comments that I have for the final version:
- It is worthwhile to clarify the difference between the representation selection setting considered in this paper, and the ones in e.g., FLAMBE, MOFFLE, and the paper of Hao et al., on sparse linear RL. I feel the two settings are quite different, and it is worth clarifying this for readers and potential new contributors to the area. To me, the main differences are: in this paper all representations satisfy realizability/linear-MDP assumptions, so all of them can yield \sqrt{T}-type regret, while one can potentially yield a much better guarantee. In the aforementioned work, only one representation can even yield \sqrt{T}-type regret, while the others are essentially useless. In particular, the assumptions here are quite a bit stronger, but of course the result is also stronger.
- On a technical level, it would be helpful to explain how the proofs here differ from the analogous results in linear bandits. What technical novelty should a reader expect here?
- Lastly, please include any details from the discussion with the reviewers into the final manuscript. In particular the empirical evaluation would be nice to add.

**Consistency Experiment:**

NeurIPS has a long history of experimentation. In 2014, NeurIPS ran an experiment in which 10% of submissions were reviewed by two independent committees to quantify the randomness in the review process. This year, we repeated a variant of this experiment to see how the quality of the review process has changed over time.  This paper was part of the experiment and was therefore assigned to two committees (consisting of reviewers, an Area Chair, and a Senior Area Chair) that reached independent decisions.  If both committees made the same recommendation, this recommendation was followed. If a single committee recommended acceptance, the paper was accepted (with the exception of a few cases in which the other committee identified what we considered a fatal flaw, e.g., an error in a key result).

This copy’s committee reached the following decision: **Accept (Poster)**

The other committee assigned to the paper recommended **Reject**.  You can find the other set of reviews, along with any follow up discussion with the authors here:
https://openreview.net/forum?id=bGVZ6_u08Jy